# EFA6B regulates a stop signal for collective invasion in breast cancer

Racha Fayad[1,5], Monserrat Vázquez Rojas[1,5], Mariagrazia Partisani[1], Pascal Finetti [2], Shiraz Dib[1], Sophie Abelanet[1], Virginie Virolle[3], Anne Farina[4], Olivier Cabaud[2], Marc Lopez [2], Daniel Birnbaum [2], François Bertucci[2], Michel Franco [1] & Frédéric Luton [1✉]

Cancer is initiated by somatic mutations in oncogenes or tumor suppressor genes. However, additional alterations provide selective advantages to the tumor cells to resist treatment and develop metastases. Their identification is of paramount importance. Reduced expression of EFA6B (Exchange Factor for ARF6, B) is associated with breast cancer of poor prognosis. Here, we report that loss of EFA6B triggers a transcriptional reprogramming of the cell-to-ECM interaction machinery and unleashes CDC42-dependent collective invasion in collagen. In xenograft experiments, MCF10 DCIS.com cells, a DCIS-to-IDC transition model, invades faster when knocked-out for EFA6B. In addition, invasive and metastatic tumors isolated from patients have lower expression of EFA6B and display gene ontology signatures identical to those of EFA6B knock-out cells. Thus, we reveal an EFA6B-regulated molecular mechanism that controls the invasive potential of mammary cells; this finding opens up avenues for the treatment of invasive breast cancer.

[1] CNRS UMR7275, Institut de Pharmacologie Moléculaire et Cellulaire (IPMC), Université Côte d'Azur, Valbonne, France. [2] INSERM U1068, Centre de Recherche en Cancérologie de Marseille, Equipe Oncologie Prédictive, Aix-Marseille Université UM105, Marseille, France. [3] CNRS UMR7277, Inserm U1091, Institut de Biologie Valrose, Université Côte D'Azur, Nice, France. [4] INSERM U1068, Centre de Recherche en Cancérologie de Marseille, ICEP Platform, Aix-Marseille Université UM105, Marseille, France. [5] These authors contributed equally: Racha Fayad, Monserrat Vázquez Rojas. ✉email: luton@ipmc.cnrs.fr

B reast cancer (BC) is a major public health issue with half a million deaths worldwide each year, essentially due to metastatic dissemination[1]. Despite extensive research, there is still no marker to predict the transition from in situ to invasive carcinoma, and treatment of metastasis remains a largely unresolved problem. There is therefore an urgent need to understand better the molecular mechanisms that support tumor invasion.

Physiological or tumor invasion of cells within the extracellular matrix (ECM) is defined according to several criteria: individual or collective migration, non-proteolytic or degradative invasion of the matrix, and structural remodeling of the ECM to facilitate migration by applying cell-generated forces at adhesion sites[2]. It has been proposed that invasion of tumor cells is due to a lack of responsiveness to stop signals provided by the ECM. A large number of studies have described the cellular and molecular mechanisms that promote or sustain invasion but much less is known about intracellular signaling pathways that restrain the invasive potential of normal epithelial cells or transduce stop signals to neoplastic cells[3–6].

We have previously reported the anti-tumor potential of the ARF6 exchange factor, EFA6B. We showed that its level of expression determines the epithelial status of mammary cells grown in 3D culture. In particular, the over-expression of EFA6B in weakly malignant tumor cells restores a normal epithelial phenotype[7,8]. In BC patients, we observed a correlation of the loss of expression of EFA6B with the metastatic Triple-Negative subtype and with a reduced survival rate in BC[7]. Others found that activation of EFA6B/ARF6 inhibits migratory and invasive properties in vitro[9]. Here, to determine the mechanism by which the loss of EFA6B might facilitate the progression of metastatic tumors, we analyzed the consequence of knocking out its gene (PSD4) in normal human mammary cells.

We found that EFA6B knock-out (KO) mammary cells undergo collective invasion in 3D-collagen. This invasion is supported by the activation of an epithelial-to-mesenchymal transition (EMT) program and the alteration of ECM interaction. EFA6B KO leads to the activation of CDC42, which in turn elicits two signaling pathways essential for invading a 3D-collagen matrix: Cdc42-MRCK-pMLC, which regulates contractility, and Cdc42-N-WASP-Arp2/3, which is required for the formation of integrin β1-based and MMP14-enriched invadopodia. Consistently, EFA6B KO in MCF10 DCIS.com cells stimulates ductal carcinoma in situ (DCIS) to invasive ductal carcinoma (IDC) transition in xenograft experiments. Further, the expression of EFA6B is lower in invasive than in in situ tumors isolated from patients, and the invasive tumors display gene ontology signatures identical to those of EFA6B knock-out cells. Thus, we reveal an EFA6B-regulated molecular mechanism that controls the invasive potential of mammary cells; this finding opens up avenues for the treatment of invasive BC.

## Results

**Loss of EFA6B in MCF10A stimulates invasion.** To address the mechanism of action of EFA6B, we have knocked-out its gene PSD4 using the CRISPR/Cas9 technology. The MCF10A human mammary cell line was used for the knock-out as it is a well-characterized normal human mammary cell line and thus enabled us to study the effect of deleting PSD4 in a non-transformed genetic background. Figure 1a shows the characterization of three homozygous (KO55, KO50, and KO2) and one heterozygous (Het2.9) KO clones, with the latter expressing half of the total levels of EFA6B compared to wild-type (WT) cells. A slight decrease (1.4 ± 0.4 fold) of ARF6 expression was observed in EFA6BKO cells (Fig. 1a, b), which was also noticed in BC patients whose EFA6B expression was decreased[7]. Notably, ARF6GTP

levels were reduced (2.5 ± 0.4 fold) indicating that EFA6B is a major ARF6-GEF in MCF10A cells (Fig. 1b). The levels of the other EFA6 and ARF proteins remained unaffected.

To assess the hallmark property of epithelial cells to polarize and self-organize in acini, the clones were placed in 3D-collagen I gels. MCF10A WT, MCF10A WT expressing a sgRNA control, and two of the WT clones isolated during the screen formed round aggregates typical of normal epithelial cells (Fig. 1c and Supplementary Fig. 1a). In contrast, all of the KO clones had outgrowths of branched structures reminiscent of collective invasion (Fig. 1c). Quantification of cell aggregates displaying at least one branched structure showed that KO clones developed four times as many membranes and cellular protrusions than WT cells (Fig. 1d). EFA6BKO cells had neither increased cell proliferation nor migratory properties (Supplementary Fig. 1b, c). Re-expression of wild-type EFA6B was sufficient to recover the formation of normal round aggregates indicating that the invasive phenotype was specifically a consequence of the loss of EFA6B (Fig. 1a, c, d). The heterozygous Het2.9 clone organized identical branched structures to the same extent as the homozygous KO clones (Fig. 1c, d). Because exogenous expression of EFA6B in KO55 cells rescued the invasive phenotype, we propose that the dominant loss-of-function of EFA6B in the heterozygous KO cells is likely due to haplo-insufficiency. To demonstrate the collective mode of invasion, we carried out time-lapse imaging of WT and EFA6BKO cell aggregates embedded in a collagen matrix (Supplementary Movies 1 and 2). While WT aggregates remained compact projecting membrane protrusions, EFA6B KO2 and KO55 cell aggregates extended massively into the matrix yet maintaining cell–cell contacts. Few single cells transiently moved away but re-established contact with cell aggregates or invading chains. Thus, depletion of EFA6B promotes collective invasion.

**Loss of EFA6B stimulates invasion of luminal and basal mammary populations.** We next asked whether the EFA6B KO has a similar impact on both luminal and basal mammary epithelial cells. We used the HMLE human epithelial population as it contains luminal progenitors and both mature luminal and basal epithelial cells. The markers EpCAM (Epithelial Cell Adhesion Molecule) and CD49f (integrin α6) are commonly used to assess mammary cell differentiation[10]. We sorted luminal progenitors (EpCAM+/CD49f+), mature luminal (EpCAM+/CD49f−), and mature basal (EpCAM−/CD49f+) cells (Fig. 2a) and immediately performed CRISPR/Cas9-mediated mutation of PSD4 in each separate population. We obtained one homozygous (KO3) and one heterozygous (Het.25) KO clones from the luminal progenitor population and one homozygous clone (KO1) from the mature basal population. No clone was obtained from the mature luminal population. EFA6B protein expression was undetectable in the homozygous KO clones, while the heterozygous clone expressed half of its corresponding WT clone (Fig. 2b). The expression of EFA6 paralogs, EFA6A and EFA6D, was unaffected. A significant reduction of ARF6 expression was observed in the Het.25 (2.05 ± 0.24 fold) and KO3 (1.89 ± 0.26 fold) clones isolated from the luminal progenitor population but not from the basal population. However, neither ARF1 nor ARF5 levels were altered. In 3D-collagen culture, HMLE WT clones formed cohesive rounded aggregates while the EFA6B homozygous and heterozygous KO clones displayed invasive cellular protrusions (Fig. 2c). Although less branched compared to MCF10A, the HMLE KO clones formed invasive aggregates (Fig. 2d). In conclusion, EFA6B is a general negative regulator of the invasive properties of epithelial mammary cells from both luminal and basal origins.

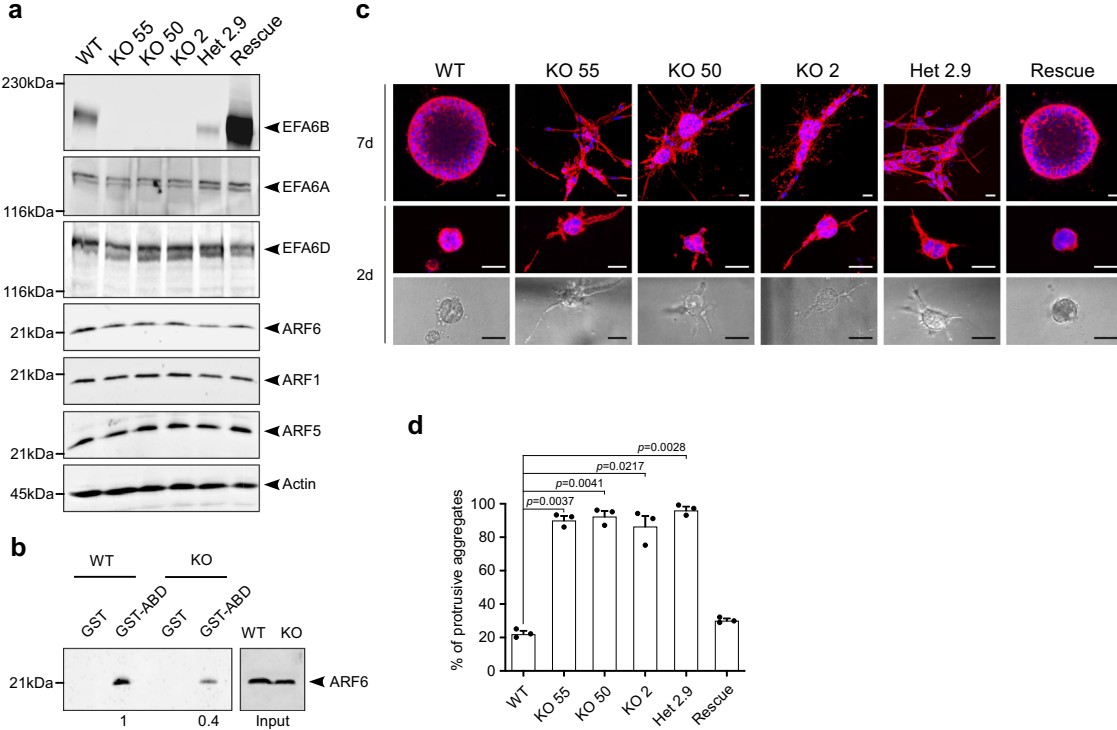

**Fig. 1 CRISPR/Cas9-mediated knock-out of the EFA6B encoding gene *PSD4* in MCF10A cells induces collective invasion in collagen I. a** The MCF10A WT, the homozygous EFA6B KO55, KO50, KO2, the heterozygous EFA6B KO2.9 (Het 2.9) and the EFA6B KO55 over-expressing EFA6B-vsvg cells were solubilized and the expression of the indicated proteins was analyzed by immunoblot. Actin served as a loading control. **b** Lysates of MCF10A WT and EFA6B KO55 cells were reacted with GST or GST-ABD (ARF6GTP-binding domain of ARHGAP10) bound to glutathione-sepharose beads. The whole lysates and bound proteins were analyzed by immunoblotting with an anti-ARF6 antibody. $N = 3$. **c** Representative images of the indicated cell aggregates placed in collagen for 7 days (upper panels) or 2 days (middle and bottom panels). The cells were processed for immunofluorescence to label the endogenous F-actin (red) and the nuclei (blue). The bottom panels are bright-field phase-contrast images of the corresponding immunofluorescence images shown in the middle panels. Scale bars 20 µm. **d** Quantification of the percentage of cell aggregates ($n = 100$) with invasive protrusions of the indicated MCF10A cell lines grown in collagen for 2 days. $N = 3$, average ± SEM, one-way ANOVA test with Dunnett's multiple comparison *p*-values. Source data are provided as a Source Data file.

**EFA6B knock-out stimulates invasion via MMP14-invadopodia**. To determine whether EFA6BKO cells acquired degradative properties, we seeded them on a fluorescent gelatin matrix. In contrast to WT cells, the KO55 clone was capable of degrading the fluorescent gelatin seen as dark spots underneath the cells (Fig. 3a, b). We also performed immunofluorescence analyses using the Col1-3/4C antibody that recognizes specifically the digested ends of collagen fibers[11]. The Col1-3/4C staining was virtually absent around the WT aggregates while a strong signal was visible along the invasive cellular protrusions extending from the KO55 aggregates (Fig. 3c, d). Thus, the KO of EFA6B enabled MCF10A cells to proteolytically cleave collagen fibers organized as a 3D-matrix. Searching for the protease responsible for the collagen degradation, we focused on MMP14 described as the main metalloprotease involved in mammary gland morphogenesis and BC metastasis[12–15]. Downregulation of MMP14 strongly inhibited collagen invasion by the EFA6BKO cells (Fig. 3e, f). Other collagenases/gelatinase such as MMP13 is not expressed in MCF10A cells, or MMP2 has an anti-invasive impact (Supplementary Fig. 2a, b). Because the degradation of the fluorescent gelatin appeared as circular spots and because WT and EFA6BKO cells expressed similar levels of MMP14 (Fig. 3g and Supplementary Fig. 6a), we searched for the formation of invadopodia enriched in MMP14. Cortactin, a well-known marker for invadopodia, was expressed at similar levels in WT and EFA6BKO cells (Supplementary Fig. 6a, KO2 cells figure). In EFA6BKO cells, staining for cortactin revealed the presence of large

structures co-stained for F-actin (Fig. 3h and Supplementary Fig. 6b). In WT cells, the cortactin positive spots, which were not as enriched in F-actin, were smaller. Further, the cortactin positive aggregates observed in EFA6BKO cells co-localized with the black spots of degraded fluorescent gelatin (Supplementary Fig. 1d). Quantification of the percentage of cells with invadopodia showed a 2-fold increase in KO cells when compared to WT cells (Fig. 3i and Supplementary Fig. 6c). We then looked for MMP14 localization in invadopodia using stable cell populations sorted to express low levels of transfected MMP14-mCherry. KO55 cells, but not WT cells, presented extensive co-localization of cortactin with MMP14-mCherry in large ventral F-actin positive structures (Fig. 3j, k). These results demonstrate that the loss of EFA6B leads to the formation of degradative MMP14-enriched invadopodia responsible for invasion within 3D-collagen.

**EFA6B knock-out promotes a change in the integrin (ITG) repertoire and stimulates the formation of ITGβ1-invadopodia**. Invasion and formation of invadopodia rely on the ITG-mediated interaction with the ECM[5]. To assess the changes promoting invasion imposed upon loss of EFA6B, we compared the gene expression profile of *PSD4* KO MCF10A cells ($N = 5$, including KO55 ($N = 3$) and KO2.9 cells ($N = 2$)) to that of *PSD4* WT MCF10A cells ($N = 3$). We identified 296 genes differentially expressed, including 173 over-expressed and 123 underexpressed in the KO cells (Supplementary Data 1). The

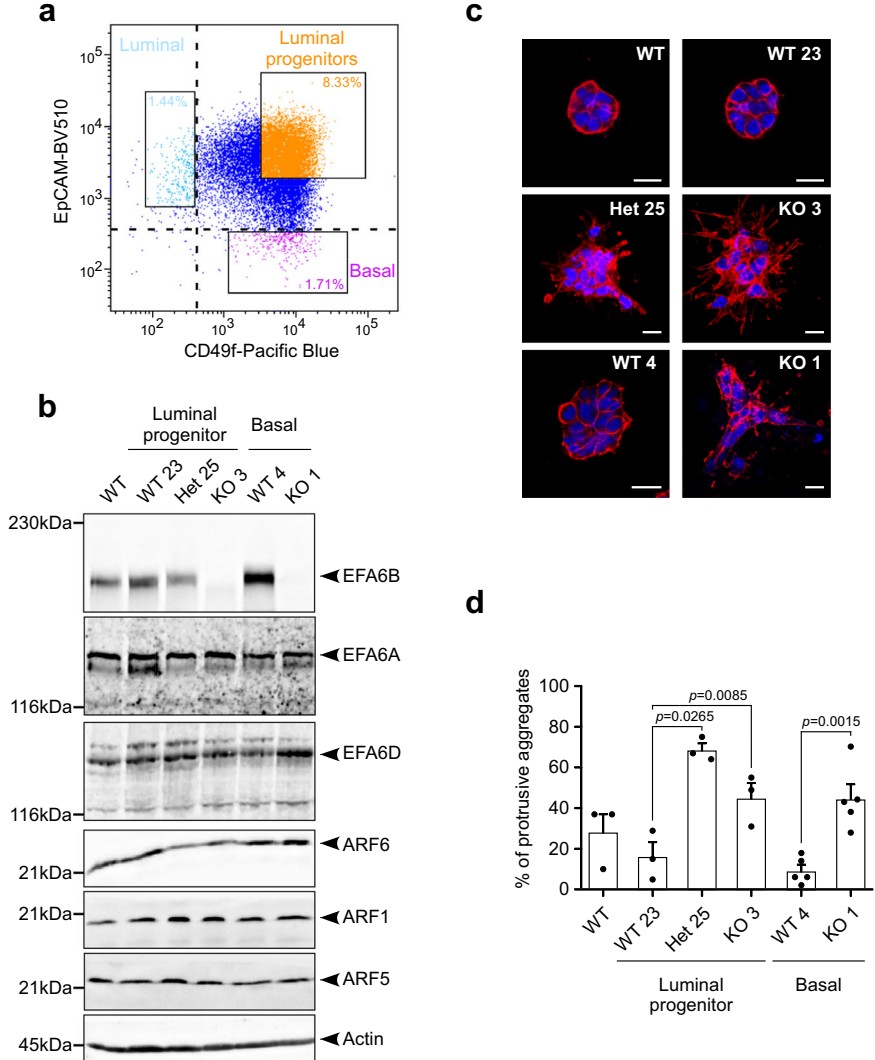

**Fig. 2 CRISPR/Cas9-mediated knock-out of the EFA6B encoding gene *PSD4* in HMLE luminal and basal populations induces collective invasion in collagen I. a** The cell surface marker EpCAM and CD49f were used to sort three epithelial cell populations including the luminal (light blue), luminal progenitors (orange) and mature basal cells (purple). These cells were immediately processed for CRISPR/Cas9-mediated *PSD4* knock-out. **b** The HMLE WT population, the luminal progenitor clone WT23, heterozygous EFA6B KO25 (Het 25), homozygous EFA6B KO3, and the mature basal clone WT4, homozygous EFA6B KO1 cells were solubilized and the expression of the indicated proteins analyzed by immunoblot. Actin served as a loading control. **c** Representative images of the indicated cells grown 5 days in collagen I and stained for F-actin (red) and the nuclei (blue). Scale bars 20 μm. **d** Quantification of the percentage of cell aggregates ($n = 100$) with invasive protrusions of the indicated cell lines grown in collagen I for 5 days. $N = 3$ for WT, WT23, Het25, KO3; $N = 5$ for WT4 and KO1, average ±SEM, paired Student's *t*-test *p*-values are versus WT23 for luminal cell lines and WT4 for basal cell lines. Source data are provided as a Source Data file.

matrisome and its receptor machineries were the main affected group of genes with up to 12% (36 genes) of the total 296 altered genes. Affected matrisome genes encoded structural glycoproteins, ECM-associated proteins, regulators and secreted factors, and ECM-receptor machinery molecules. Consistently, many gene ontologies associated with the 296-gene list were related to ECM, cell–cell signaling, cell–cell adhesion, cell-adhesion (to substratum), and EMT (Supplementary Data 2). GSEA of a "Matrisome+receptors" gene set (including the KEGG gene sets Focal Adhesion (FA), ECM-receptor interaction (ECM), and the human matrisome database) confirmed such over-representation of genes (Supplementary Fig. 2). Thus, in response to *PSD4* mutation, the cells have modified their molecular ECL composition and the expression of corresponding receptors.

We then explored the protein expression level of the major ITGs that regulate mammary morphogenesis. We found that KO55 cells have a reduction of subunits α6 (50%) and β4 (36%) expression, while subunits α2, α3, and β1 were unchanged (Fig. 4a–c). To determine which ITGs were required for EFA6BKO-mediated invasion, we quantified the number of invasive aggregates grown in gels containing control or ITG-blocking antibodies. Antibodies against α6, β4, or α3 had no or little effect. In contrast, antibodies against α2 and β1 strongly blocked cell invasion suggesting that the ITGα2β1 is required for EFA6BKO cells invasion (Fig. 4d). Since the ITGβ1 has been involved in invadopodia regulation and MMP14 enrichment[16], we assessed whether β1 was concentrated at invadopodia. We found a significant co-localization of β1 with MMP14-mCherry in large ventral F-actin positive structures in KO55 but not in MCF10A WT cells (Fig. 4e, f). Thus, the absence of EFA6B induces a modification of the matrisome and ITG repertoire along with the formation of degradative ITGβ1-based

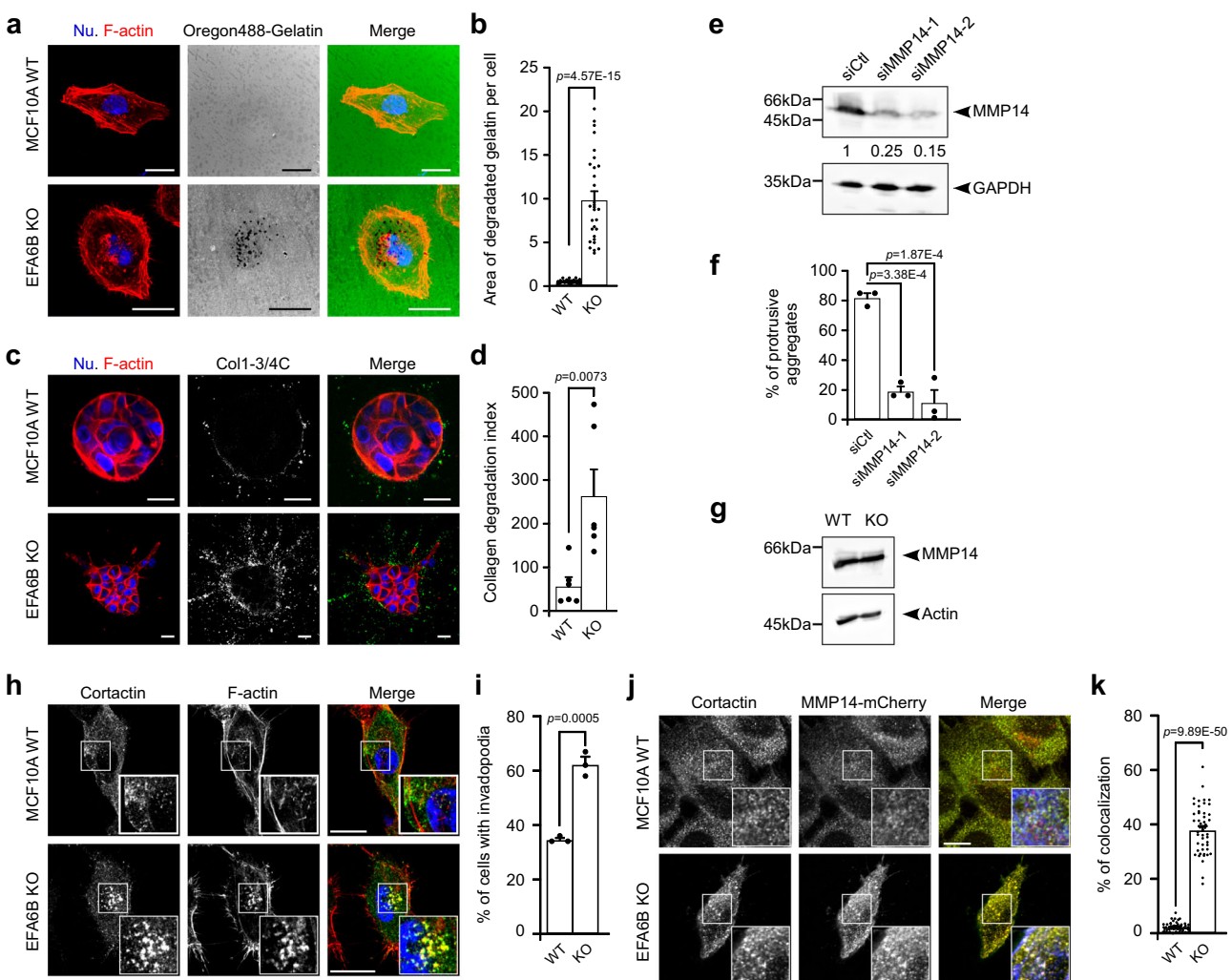

**Fig. 3 EFA6B knock-out stimulates matrices degradation and invasion in an MMP14-dependent manner. a** Representative images of MCF10A WT and EFA6B KO55 cells placed on Oregon488-gelatin (green)-coated coverslips and stained for F-actin (red) and nuclei (Nu., blue). Areas devoid of fluorescent signal indicate degradation of the fluorescent gelatin. Scale bars 20 μm. **b** Quantification of the gelatin degradation. Values are the mean percentage of degradation area per cell area ± SEM. $n = 10$, $N = 3$, Student's $t$-test $p$-value. **c** Representative images of MCF10A WT and EFA6B KO55 cells grown in collagen I for 3 days and stained for cleaved collagen I with the Col1-$^{3/4}$C antibody (white in middle panel and green in left merge panel), for F-actin (red) and nuclei (blue). Scale bars 20 μm. **d** Quantification of collagen degradation. Values are mean degradation index ± SEM. $n = 3$ aggregates of 30 cells, $N = 2$, Student's $t$-test $p$-value. **e** MCF10A WT and EFA6B KO55 cells were transfected with siRNA control or directed against MMP14. 48 h post-transfection the expression of MMP14 was analyzed by immunoblot. GAPDH served as a loading control. $N = 3$. **f** Quantification of the percentage of cell aggregates ($n = 100$) with invasive protrusions of the indicated cell lines grown in collagen I for 2 days. $N = 3$, one-way ANOVA test with Dunnett's multiple comparison $p$-values. **g** Expression of MMP14 analyzed by immunoblot in MCF10A WT and EFA6B KO55 cells. GAPDH served as a loading control. $N = 4$. **h** Representative images of the indicated cells grown 2 days on collagen I-coated coverslips stained for cortactin (green), F-actin (red), and nuclei (blue). The large inset is a 2× zoom-in image of the indicated area. Scale bars 20 μm. **i** Quantification of the percentage of cells ($n = 100$) displaying invadopodia, $N = 3$, average ± SEM, Student's $t$-test $p$-value. **j** Representative images of the indicated cells grown 2 days on collagen I-coated coverslips stained for cortactin (green), MMP14-mCherry (red), and F-actin (blue). Co-localization of all three markers appears in white. The large inset is a 2× zoom-in image of the indicated area. Scale bars 20 μm. **k** Quantification of the percentage of cortactin co-localized with MMP14-mCherry. A total of 52 WT cells and 48 KO55 cells from three independent experiments were analyzed, Student's $t$-test $p$-value. Source data are provided as a Source Data file.

invadopodia, all of which contribute to the collective invasion of the EFA6BKO cells.

**EFA6B knock-out induces the expression of EMT transcription factors that promote collective invasion of MCF10A and HMLE cells in collagen.** EMT is a major molecular program promoting collective invasion believed to provide cells with migratory and degradative advantages[5,17]. We observed by immunoblot and immunofluorescence that the MCF10A EFA6BKO cells had reduced levels of E-cadherin together with an increased expression of N-cadherin (Fig. 5a and Supplementary

Fig. 3a, f). Knowing that EFA6B is a tight junction (TJ) regulator, we also looked at TJ molecules and found that CLDN1 and CLDN3 were strongly reduced in all clones and occludin decreased in 3 out of 4 clones (Fig. 5a). We assessed the cell-to-cell affinity by using the hanging-drop assay. MCF10A WT cells formed compact monolayers while KO55 cells formed lacy monolayers indicative of loose cell–cell contacts (Supplementary Fig. 3b). In addition, transcriptomic analyses of the KO55 and Het2.9 clones showed a significant alteration of the expression of the Gene Ontology "cell–cell adhesion" signature (Supplementary Fig. 2 and Supplementary Data 2). These results support a mode

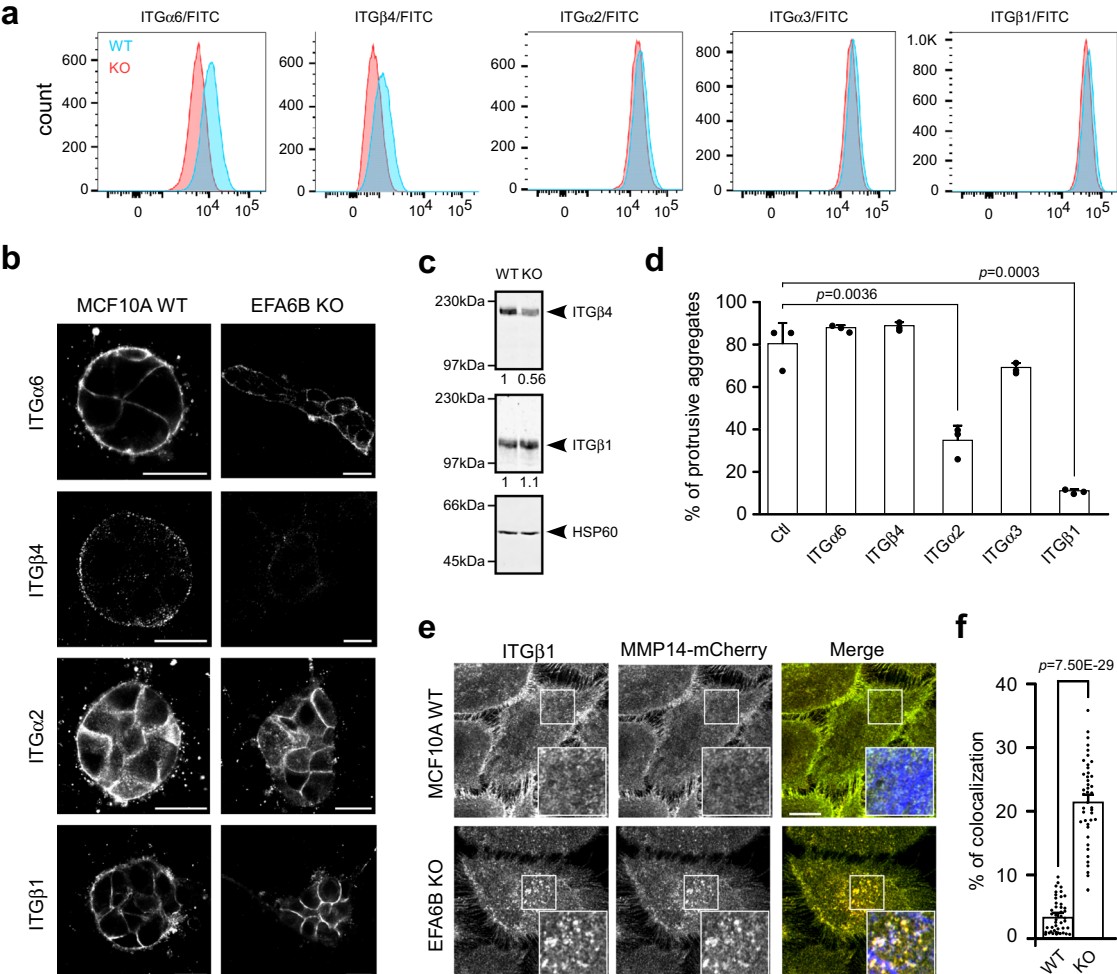

**Fig. 4 EFA6B knock-out promotes a change in the ITG repertoire and stimulates the formation of ITGβ1-based invadopodia. a** Cell surface expression of ITG molecules in MCF10A WT (blue) and EFA6B KO55 (red) cells analyzed by FACS. **b** Representative images of the indicated spheroids grown 2 days in collagen I stained for the indicated ITG. Scale bars 20 μm. **c** Expression of ITGβ1 and ITGβ4 analyzed by immunoblot in MCF10A WT and EFA6B KO55 cells. HSP60 served as a loading control. $N = 3$. **d** Quantification of MCF10A WT and EFA6B KO55 cell aggregates with invasive protrusions incubated in the presence of the control pre-immune serum (Ctl) or the indicated anti-ITG (α-ITG) antibodies for 2 days. Values are percentages of total cell aggregates ± SEM. 300 cell aggregates were analyzed for each cell population in three independent experiments, one-way ANOVA test with Dunnett's multiple comparison p-values. **e** Representative images of the indicated cells grown 2 days on collagen I-coated coverslips stained for ITGβ1 (green), MMP14-mCherry (red), and F-actin (blue). Co-localization of all three markers appears in white. The large inset is a 2× zoom-in image of the indicated area. Scale bars 20 μm. **f** Quantification of the percentage of ITGβ1 co-localized with MMP14-mCherry. A total of 45 WT cells and 42 KO55 cells from three independent experiments were analyzed, Student's t-test p-value. Source data are provided as a Source Data file.

of collective invasion whereby reduced cell–cell adhesion facilitates pro-migratory cell movements while maintaining cell–cell contacts.

Further analysis by RT-qPCR confirmed the E/N-cadherin switch, the decrease of TJ markers, and of CK14 whose downregulation was recently shown to mark an advanced mesenchymal state in melanoma and breast tumors[18], and also the decrease of the mammary differentiation marker *CD49f* (Fig. 5b). We found an overall decrease of EpCAM and CD49f ($66.9 \pm 7.2\%$ and $41.4 \pm 9.6\%$, respectively), together with the emergence of a new EpCAM$^{-/low}$ and CD49f$^{low}$ population (11.7% of total cells, red gate), suggestive of a loss of epithelial identity of the KO55 cells (Supplementary Fig. 3c). We also looked at whether the re-expression of EFA6B could revert the EMT. We found that it could revert the loss of expression of the epithelial markers E-cadherin and Cld1, however, the levels of vimentin remained unaffected and those of N-cadherin even increased, yet the cells had retrieved their epithelial

organization in 3D-collagen and were no longer invasive. These observations indicate that a partial reversal of EMT is sufficient to restore a non-invasive epithelial phenotype (Supplementary Fig. 4a).

We similarly analyzed the EMT status of the HMLE EFA6BKO clones. In contrast to MCF10 EFA6BKO cells, we did not notice a change in E- or N-cadherin expression. However, we found a strong increase of vimentin and a slight but consistent decrease of CLDN3 expression (Fig. 5c and Supplementary Fig. 3f). Because of the well-documented variable outcome of EMT[17], a more recent definition of EMT relies on the turn-on of EMT-activating transcription factors (EMT-TFs)[19]. We found a significant increase of SNAIL1, TWIST1, and ZEB1, and to a lower extent of SNAIL2 and TWIST2 expression in the KO55 cells (Fig. 5b). In HMLE, SNAIL1 was the major EMT-TF commonly up-regulated in all three KO clones (Fig. 5d and Supplementary Fig. 3d). Hence, depending on the cell lines, the elicited EMT-TFs and the EMT program triggered by EFA6BKO are variable, yet altogether

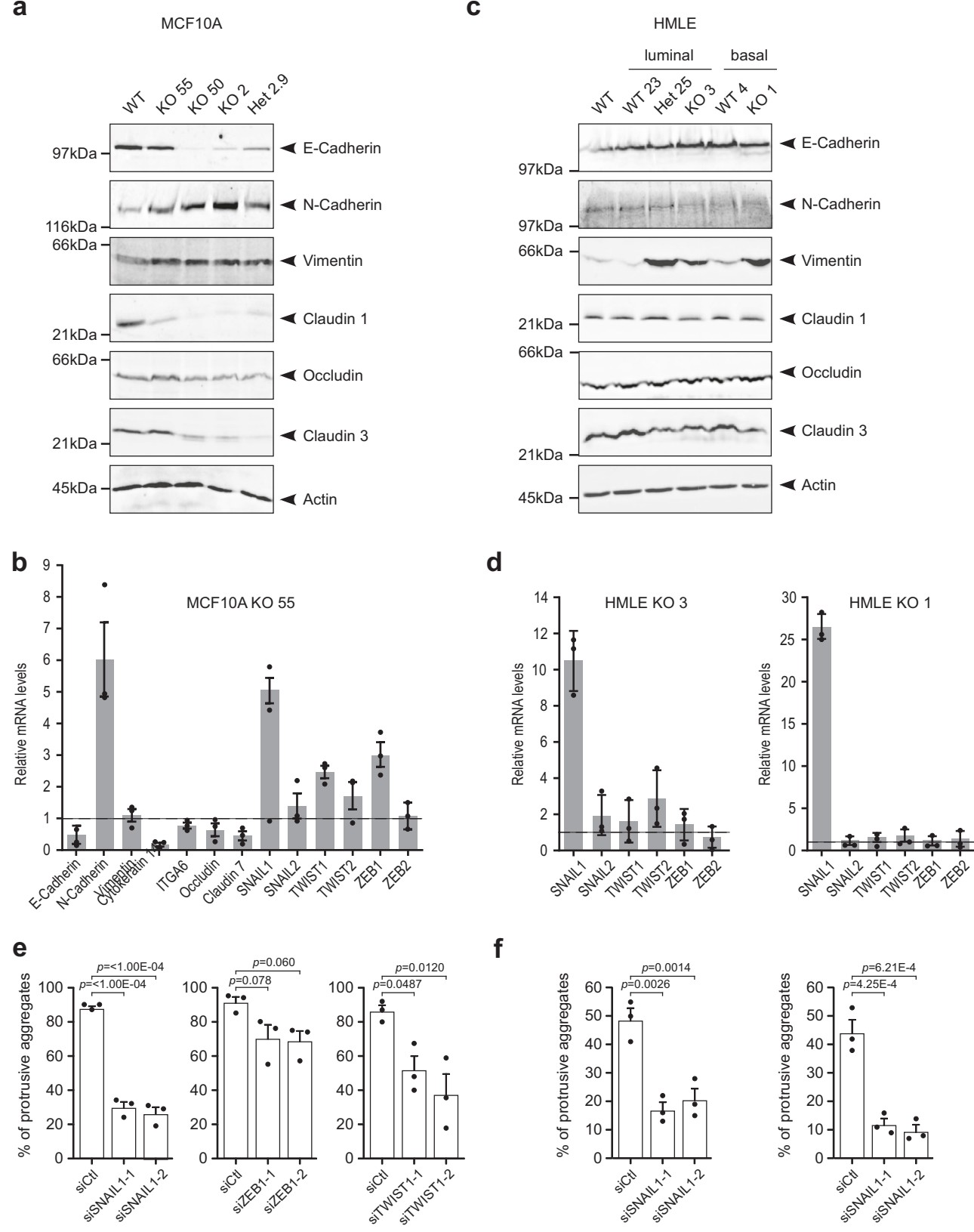

our results indicate that the loss of EFA6B engages epithelial cells into EMT.

Activation of EMT-TFs was shown to favor dissemination by increasing invasion[5,17]. We found that siRNA-mediated down-regulation of SNAIL1 in KO55 cells blocked invasion, while that of TWIST1 only reduced it and that of ZEB1 had no effect (Fig. 5e and Supplementary Fig. 3e). Similarly, depletion of SNAIL1 in both HMLE KO clones blocked invasion (Fig. 5f and Supplementary Fig. 3e), indicating that SNAIL1 is the major EMT-TF required for collagen invasion of EFA6BKO cells.

**Fig. 5 EFA6B knock-out induces the expression of EMT transcription factors that promote collective invasion of MCF10A and HMLE WT cells in collagen I. a** The MCF10A WT, the homozygous EFA6B KO55, KO50, KO2, and the heterozygous EFA6B KO2.9 (Het2.9) cells were solubilized and the expression of the indicated proteins analyzed by immunoblot. Actin served as a loading control. $N = 3$. Quantification is shown in Supplementary Fig. 3f. **b** Expression of EMT-associated genes by qPCR analysis in EFA6B KO55 cells normalized to MCF10A WT. $N = 3$, average ± SEM. **c** The HMLE WT population, the luminal progenitor clone WT23, heterozygous KO25, homozygous KO3, and the mature basal clone WT4, homozygous KO1 cells were solubilized and the expression of the indicated proteins analyzed by immunoblot. Actin served as a loading control. $N = 3$. Quantification is shown in Supplementary Fig. 3f. **d** Expression of EMT-TF genes by qPCR analysis in EFA6B KO3 and KO1 cells normalized to their respective HMLE control cells WT23 and WT4. $N = 3$, average ± SD. **e** Quantification of the percentage of aggregates ($n = 100$) with invasive protrusions of MCF10A KO55 cells grown in collagen I for 2 days after transfection with the indicated siRNAs. $N = 3$, average ± SEM, one-way ANOVA test with Dunnett's multiple comparison $p$-values. **f** Quantification of the percentage of aggregates ($n = 100$) with invasive protrusions of HMLE KO3 (left) and KO1 (right) cells grown in collagen I for 2 days after transfection with the SNAIL1 targeted siRNAs. $N = 3$, average ± SEM, one-way ANOVA test with Dunnett's multiple comparison $p$-values. Source data are provided as a Source Data file.

EMT has been found to be associated with a gain in stemness properties[20,21]. Analyses of the MCF10A clones failed to detect the formation of any mammosphere and showed no gain in the mammary stem cell characteristics (Supplementary Fig. 4b–d).

We concluded that the knock-out of EFA6B in mammary cells induces the expression of EMT-TFs triggering an EMT program that promotes dedifferentiation and facilitates collective invasion.

**EFA6B knock-out stimulates cellular contractility and invasion through CDC42.** ECM remodeling and cellular contractility are important regulators of the formation of invadopodia[22,23]. We first studied the impact of EFA6B KO on cell contractility by using a collagen gel contraction assay. KO55 and KO2 cells were at least twice more contractile than their WT counterparts (Fig. 6a and Supplementary Fig. 6d). Cell contractility is dependent upon the activation of the myosin II through the phosphorylation of the myosin light chain (MLC)[24]. KO55 and KO2 cells presented higher levels of pMLC than control cells (Fig. 6b and Supplementary Fig. 6e). We then looked at the organization of the fibrillary collagen surrounding the cell aggregates by reflectance (Fig. 6c). We could not distinguish any particular orientation of the collagen fibers, which were randomly distributed around the WT cell aggregates. In contrast, reflectance signals showed a radial distribution of aligned collagen fibers in KO55 invasive cells. In addition, the collagen fibers were in perfect alignment with membrane filopodial structures extended from the cells (Fig. 6c). This observation indicated that KO55 cells have the capacity to remodel the collagen fibers into tracks to facilitate their invasion. Small G proteins of the RHO family are known to control cell contractility[25]. In both EFA6B KO55 and KO2 cells, we found that the downregulation of CDC42, but not RHOA, RHOC, or RAC1, altered the contractility of (Fig. 6a, d and Supplementary Fig. 6d). Next, we observed that the downregulation of CDC42 was the most effective at blocking invasion (Fig. 6d, e and Supplementary Figs. 5b, c, 6f). It also hampered the formation of invadopodia and the extension of short filopodia (Fig. 6f, g and Supplementary Figs. 5a, 6h). Although depletion of RHOA, RHOC, or both promoted the extension of large membrane protrusions and depletion of RAC1 induced branched single cell-wide elongated chains, neither condition blocked efficiently invasion (Fig. 6d, e and Supplementary Fig. 5b). In agreement with the functional assays, we found an upregulation of activated CDC42 (2.8 ± 0.8), but not of RHOA (1.1 ± 0.1), RHOC (1.08 ± 0.2) nor RAC1 (0.97 ± 0.10) (Fig. 6h). Overall, these observations demonstrated that EFA6B KO leads to the activation of CDC42, which regulates cell contractility, formation of filopodia and invadopodia, collagen remodeling, and collective invasion in 3D-collagen.

**EFA6B knock-out stimulates cellular contractility and invasion through CDC42-MRCK-pMLC and CDC42-N-WASP-ARP2/3 pathways.** CDC42 was shown to regulate contractility through the recruitment of kinases that phosphorylate MLC[26,27], as well as to control invadopodia formation by activating the ARP2/3 complex through N-WASP[28]. Downstream of CDC42, the myotonic dystrophy-related CDC42-binding kinases (MRCK) phosphorylate directly or indirectly MLC[29]. SiRNA experiments showed that the combined downregulation of MRCKα/β in KO55 cells reduced both MLC phosphorylation and cell invasion in collagen (Fig. 7a). RHOA activates the RHO-associated protein kinases ROCK1 and ROCK2 to phosphorylate MLC[30,31]. Knockdown of both ROCK1/2 had no effect on either phosphorylation of MLC or invasion (Fig. 7b). Thus, in KO55 cells activated CDC42 controls phosphorylation of MLC and invasion in an MRCKα/β-dependent manner. Further, siRNA downregulation of N-WASP and ARP3, both proteins similarly expressed in WT and EFA6BKO cells inhibited invasion (Fig. 7c, d and Supplementary Fig. 6a). Altogether, these results unraveled a pathway regulated by EFA6B, which controls the activation of CDC42 and two of its effector functions: contractility through MRCKα/β phosphorylation of MLC and invasion through N-WASP and ARP2/3.

**EFA6B knock-out accelerates invasion of DCIS.com xenografts.** To assess in vivo the impact of the loss of EFA6B, we have used the MCF10DCIS.com xenograft model that recapitulates the transition from ductal carcinoma in situ (DCIS) to invasive ductal carcinoma (IDC)[32]. Tumor samples were analyzed after HES coloration and staining for the myoepithelial marker p63 to help distinguish DCIS from IDC (Fig. 8a and Supplementary Fig. 7a). At the onset, tumor cells within DCIS ducts are p63-negative and surrounded by a layer of myoepithelial tumor cells expressing p63. By week 3, xenografts from control DCIS.com displayed some signs of tumor infiltration, followed by clear invasion at week 4, mixed DCIS and IDC phenotypes at week 5, and full IDC at week 6. EFA6BKO xenografts presented abnormal ductal phenotype and infiltrating foci as early as at week 2 where the percentage of p63-positive cells in EFA6BKO xenografts was increased. Mixed IDC and DCIS phenotypes at week 3 and full IDC at week 4 (Fig. 8a) were followed by an increased EFA6BKO tumor growth (Fig. 8b). Further work is needed to define the mechanisms of invasion of the DCIS.com KOEFA6B cells, however, these results indicate that reduction of EFA6B levels causes accelerated foci infiltration in vivo.

**EFA6B expression is downregulated in human clinical BC samples endowed with invasive properties.** Our results demonstrate that the reduction of EFA6B expression in normal mammary cells is sufficient to promote the acquisition of invasive properties. Within transformed cells, the loss of EFA6B could provide a pro-invasive advantage favoring the transition from in situ to the invasive lesion. To assess the impact of EFA6B

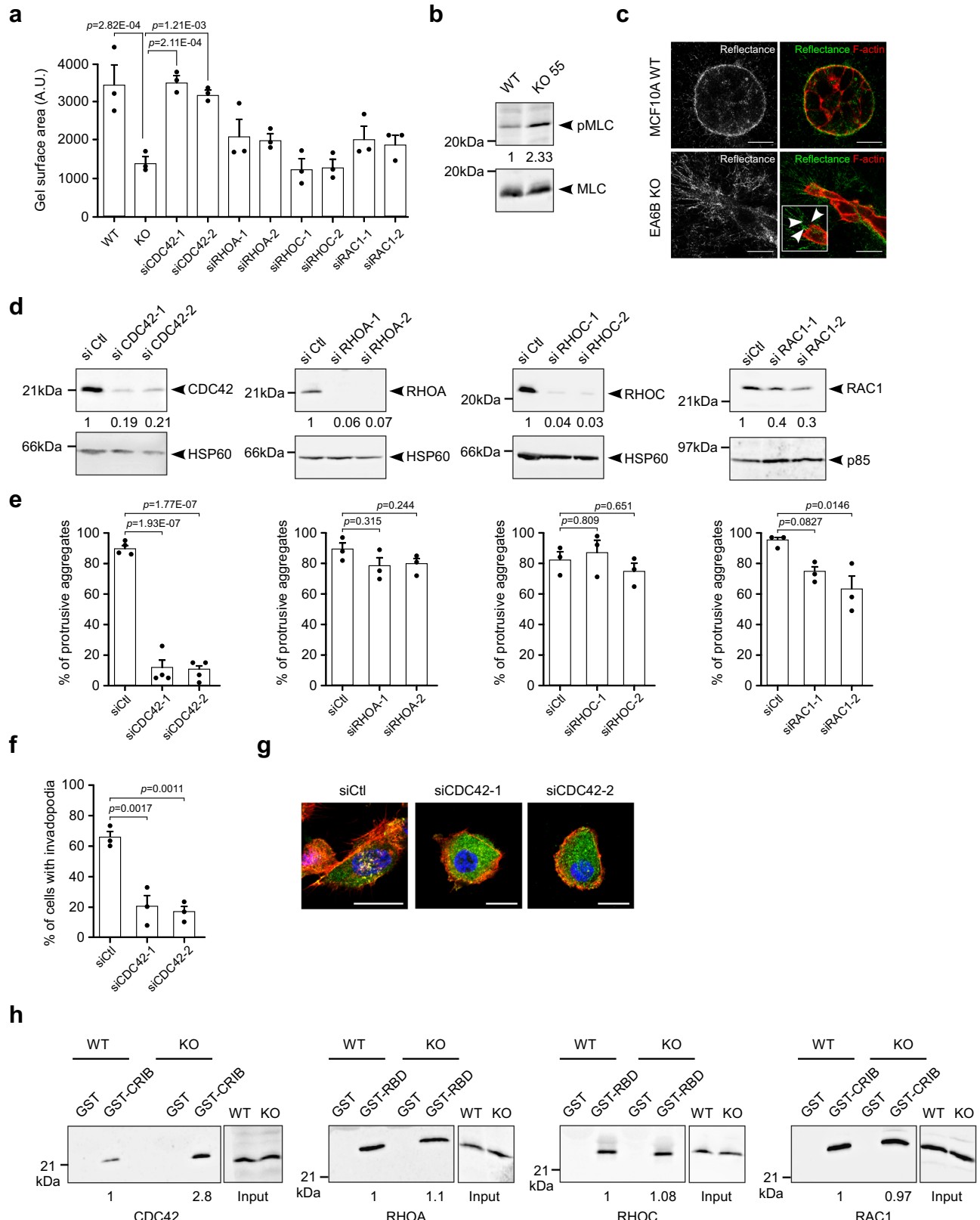

downregulation in BC progression, we compared the *PSD4* mRNA expression levels in unpaired clinical samples of DCIS and IDC[33,34]. We found a significant, small reduction of *PSD4* expression, but not of the other family members in IDC compared to DCIS (Fig. 9a). We then analyzed separately tumor epithelium and adjacent stroma prepared by laser-capture microdissection for each of the 22 unpaired DCIS ($N = 11$) and IDC ($N = 11$)[33]. A significant reduction of *PSD4* levels was observed only in the epithelial compartment (Fig. 9b), which corroborates our results showing that the loss of EFA6B in cells of epithelial origin generates invasive mammary cells (Fig. 2). Interestingly, several gene ontologies associated with our 296-

**Fig. 6 EFA6B knock-out stimulates cellular contractility and invasion through CDC42. a** Quantification of the contractility of MCF10A WT (WT) and EFA6B KO55 (KO) cells transfected with a control siRNA, and EFA6B KO55 cells transfected with the indicated siRNA evaluated by a collagen gel contraction assay. Values are the mean surface area of the collagen gel ± SEM. $N = 3$, one-way ANOVA test with Dunnett's multiple comparison $p$-values calculated versus KO55. **b** Expression of pMLC (phospho-myosin light chain) and total MLC (myosin light chain) analyzed by immunoblot in MCF10A WT and EFA6B KO55 cells, $N = 4$. **c** Representative images of the MCF10A WT and EFA6B KO55 spheroids embedded 2 days in collagen I stained for F-actin (red). The organization of the collagen I fibers surrounding the cell aggregates were imaged by confocal reflectance microscopy (green). The large inset is a 2× zoom-in image of the leader cell. Arrowheads point to thin membrane extensions co-localized with collagen fibers. Scale bars 20 μm. **d** Two days post-transfection of EFA6B KO55 cells with the indicated siRNAs, the expression of the corresponding protein was analyzed by immunoblot and quantified, $N = 3$. HSP60 and p85 served as loading controls. **e** Quantification of the percentage of aggregates ($n = 100$) with invasive protrusions of EFA6B KO55 cells grown in collagen I for 2 days after transfection with the indicated siRNAs. $N = 4$ for siCDC42 and $N = 3$ for siRHOA, siRHOC, siRAC, average ± SEM, one-way ANOVA test with Dunnett's multiple comparison $p$-values. **f** Quantification of the percentage of invadopodia in EFA6B KO55 cells ($n = 30$) grown in collagen I for 2 days after transfection with CDC42 targeted siRNAs. $N = 3$, average ±SEM, one-way ANOVA test with Dunnett's multiple comparison $p$-values. **g** Representative images of EFA6B KO55 cells transfected with the indicated siRNAs and stained for cortactin (green), F-actin (red), and nuclei (blue). Scale bars 20 μm. **h** Lysates of MCF10A WT and EFA6B KO55 cells were reacted with GST, GST-CRIB (CDC42GTP- and RAC1GTP-interacting domain of PAK) or GST-RBD (RHOAGTP- and RHOCGTP-binding domain of rhotekin) prebound to glutathione-sepharose beads. The whole lysates and bound proteins were analyzed by immunoblot with the indicated antibodies, $N = 3$. Source data are provided as a Source Data file.

gene signature and related to ECM organization, collective migration and EMT (Fig. 9c), were also associated with ontologies associated with the 131 and 120 genes that we found as differentially expressed in IDC versus DCIS in the Lee's and Knudsen's data sets, respectively[33,34]. Thus, ontologies associated with *PSD4* knock-out in MCF10A cells correlate with those of the progression of DCIS to IDC in patients. We also compared the *PSD4* mRNA expression in metastatic samples versus the paired primary BC by taking advantage of a publicly available series of 29 matched metastasis/primary cancer pairs[35,36]. We found a significant reduction of *PSD4* expression in the metastatic samples (Fig. 9d), further corroborating the role of EFA6B loss in tumor invasion. The same comparative analysis of *PSD4* expression between DCIS and IDC, and then between primary tumors and paired metastases showed similar results to the Triple-Negative (TNBC) subtype (Supplementary Fig. 7b–d): higher expression of *PSD4* in DCIS than in IDC, and higher expression in primary tumors than in metastases.

Finally, to confirm and extend our previous results[7] on a large and independent series, we searched for a correlation between *PSD4* mRNA expression and the clinico-pathological features of a publicly available series of 3613 invasive primary BC (Supplementary Table 1). From this cohort, a total of 306 tumors showed a two-fold or greater downregulation of *PSD4*, using normal breast tissue as the standard. *PSD4* downregulation was associated (Fisher's exact test; Supplementary Table 2) with younger patients' age, higher pathological grade and tumor size, ductal type, and higher frequency of TN subtype ($p < 0.001$), and shorter disease-free survival (DFS). Within the 3353 non-stage IV patients with follow-up available, the 5-year DFS was 82% (95 CI% 80–83%) for the whole population, and 69% (95 CI%, 62–76%) and 83% (95 CI% 80–84%) in cases of downregulation and no downregulation, respectively ($p = 5.71E-04$, log-rank test; Supplementary Fig. 7e).

We conclude that loss of EFA6B endows epithelial mammary cells with molecular characteristics of human invasive ductal carcinoma and lowers the 5-year DFS.

## Discussion

The invasive process occurring during embryonic development or cancer is believed to be associated with the activation of an EMT program. EMT is not a uniform program executed by a unique signaling pathway, and thus the differences between MCF10A and HMLE EFA6BKO cells likely arise from intrinsic properties to respond to EFA6B loss and carry on EMT. How the loss of EFA6B induces EMT is an open question. It could be through disassembling cell–cell junction and release of associated

transcription factors[37–41], loss of the permeability barrier giving growth factors access to their receptors[42–44], or through the remodeling of the ECM that in turn could stimulate EMT[22,45]. These scenarios could also explain the various degrees of EMT observed in between MCF10A clones. We remarked that the basal and luminal progenitor HMLE KO clones were all invasive, and that regardless of their basal or luminal origin, they displayed similar EMT profiles suggesting that luminal cells acquired at least in part basal characteristics. Thus, EFA6B depletion in both luminal and basal cells induces EMT and stimulates collective invasion.

Interestingly, the fact that the EFA6BKO cells undergo EMT, display thin filopodia at the front of the leader cells, assemble MMP14 invadopodia, and sustain invasive protrusions in collagen I is reminiscent of the mechanism of invasion by cancer cells. Indeed, the EMT program bestows carcinoma cells many of the attributes associated with invasive malignant cells[5,17]. Furthermore, collective invasion during mammary tubulogenesis occurs without actin-based protrusions extending into ECM[46]. Also, MMP14 is not expressed by the normal epithelial cells during mammary gland branching but is expressed by the invading BC cells[13]. Finally, normal cells would only invade collagen transiently before producing their basement membrane that blocks invasion[6]. Thus, we conclude that the collective invasion of EFA6B KO normal mammary cells models cancer invasion rather than normal mammary branching.

The depletion of EFA6B led to strong activation of CDC42 but not of RAC1, RHOA, nor RHOC, although they have been found in other studies to be involved in invasion and contractility[47,48]. One cannot exclude the possibility that a small fraction of other small G proteins may be activated within discrete cell area and contribute to invasion in our cell model; however, depletion of CDC42 was the most inhibitory of all molecules we have tested. In vitro and in vivo studies have shown important roles for CDC42 in regulating diverse cellular processes such as cell cycle progression and mitosis, polarity, survival, differentiation, and stem cell function[49]. In the mammary gland, transgenic overexpression of CDC42 produced hyperbranched ductal trees with abnormal acini[50], and in breast tumors, CDC42 is often found hyperactivated or over-expressed[51,52]. However, in tumor patients activating mutations in CDC42 have not been found[53], which suggests that its regulatory pathways must be altered. We propose that the loss of expression of EFA6B interferes with CDC42 regulation contributing to its hyperactivation in BC.

From in vitro studies, CDC42 is a known pro-invasive protein that can induce the formation of filopodia, invadopodia and can

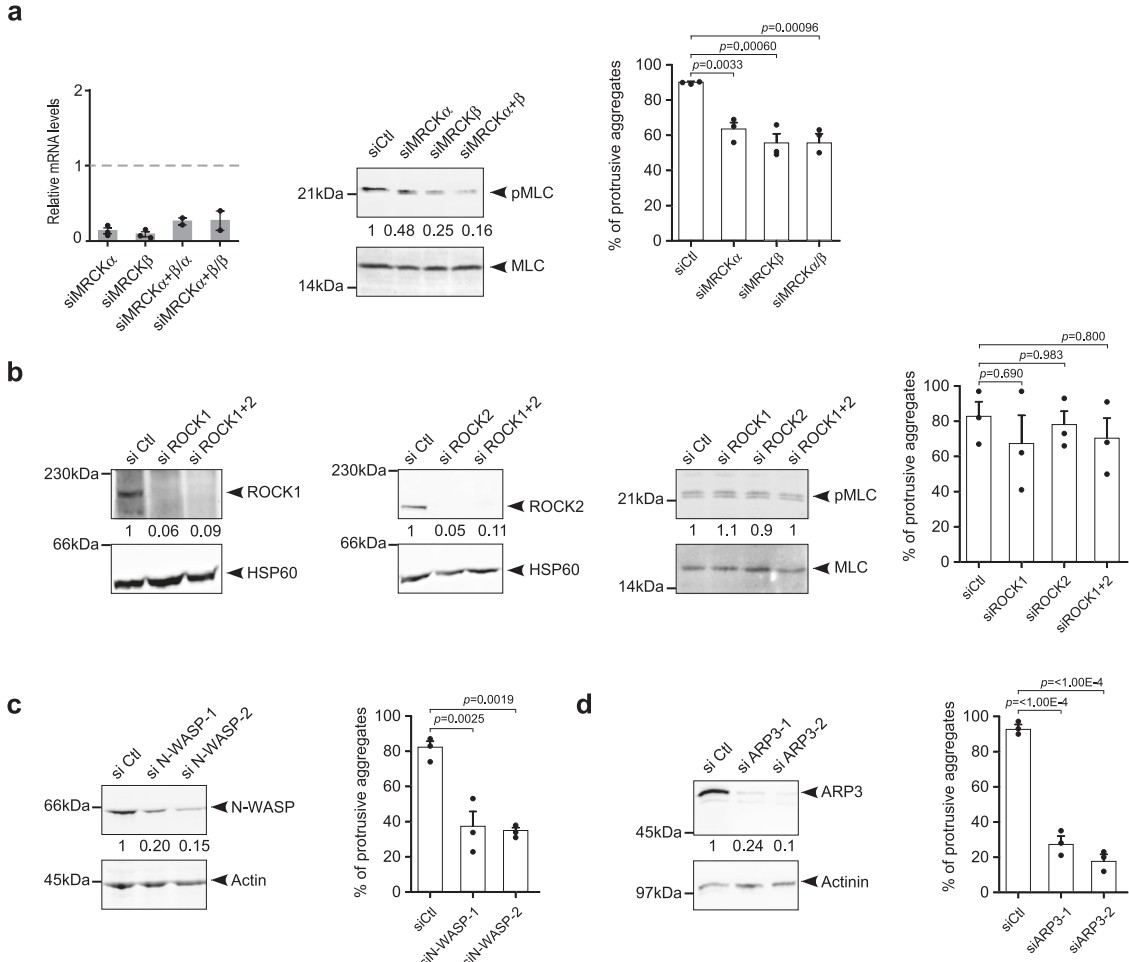

**Fig. 7 EFA6B knock-out stimulates cellular contractility and invasion through two CDC42-dependent signaling pathways: CDC42-MRCK-pMLC and CDC42-N-WASP-ARP2/3. a** Left: expression of *MRCKα* and *MRCKβ* (myotonic dystrophy kinase-related Cdc42-binding kinase) genes in EFA6B KO55 cells analyzed by qPCR 2 days post-transfection with the indicated siRNAs. siMRCKα+β/α and siMRCKα+β/β indicate the gene expression level of MRCKα or MRCKβ after transfection with siRNAs against both MRCKs, $N = 3$ for single siRNA and $N = 2$ for combined siRNA. Middle: expression of pMLC and total MLC analyzed by immunoblot 2 days post-transfection of EFA6B KO55 cells with the indicated siRNAs. Right: quantification of the percentage of aggregates with invasive protrusions of EFA6B KO55 cells grown in collagen I for 2 days post-transfection with the indicated siRNAs. $N = 3$, average ±SEM, one-way ANOVA test with Dunnett's multiple comparison *p*-values. **b** Expression of the indicated proteins analyzed by immunoblot two days post-transfection of EFA6B KO55 cells with the indicated siRNAs. ROCK: Rho-associated protein kinase. HSP60 served as a loading control. Right panel: quantification of the percentage of aggregates with invasive protrusions of EFA6B KO55 cells grown in collagen I for 2 days after transfection with the indicated siRNAs. $N = 3$, average ± SEM, one-way ANOVA test with Dunnett's multiple comparison *p*-values. **c** Left: expression of N-WASP (Neural Wiskott–Aldrich Syndrome Protein) analyzed by immunoblot two days post-transfection of EFA6B KO55 cells with N-WASP targeted siRNAs. Actin served as a loading control. Right: quantification of the percentage of aggregates ($n = 100$) with invasive protrusions of EFA6B KO55 cells grown in collagen I for 2 days after transfection with N-WASP-directed siRNAs. $N = 3$, average ± SEM, one-way ANOVA test with Dunnett's multiple comparison *p*-values. **d** Left: expression of ARP3 (Actin Related Protein 3) analyzed by immunoblot 2 days post-transfection of EFA6B KO55 cells with ARP3-targeted siRNAs. Actinin served as a loading control. Right: quantification of the percentage of aggregates ($n = 100$) with invasive protrusions of EFA6B KO55 cells grown in collagen I for 2 days after transfection with ARP3-targeted siRNAs. $N = 3$, average ± SEM, one-way ANOVA test with Dunnett's multiple comparison *p*-values. Source data are provided as a Source Data file.

regulate cellular contractility[49]. Consistent with this, we found that knock-down of CDC42 abrogates contractility and invasion of the EFA6BKO cells. In contrast, the main RHOA/ROCK contractility pathway did not appear to have any role. Contractility is reflected by the formation of linearized bundles of collagen extending from the filopodia of invasive EFA6BKO cells. These collagen bundles were shown to form tracts that facilitate tumor cells migration[22,54]. Cell contractility stiffens the matrix, and thus stimulates the formation of invadopodia and secretion of MMPs[16,23,55], advocating that in EFA6BKO cells CDC42 activation facilitates the formation of invadopodia, at least in part, by increasing cell contractility. The increased contractility of

EFA6BKO cells may also be the consequence of an alteration of the matrisome that might have strengthened the ECM rigidity[22].

Further, CDC42 has a direct role in invadopodia formation[49,56] and maturation by maintaining the activation of N-WASP-ARP2/3 complex, which in turn nucleates actin polymerization[57], and the recruitment of MMP14 to invadopodia[58]. Since the invasive nature of the EFA6BKO cells is dependent on N-WASP, ARP2/3, and MMP14, we propose that CDC42 is the major effector of this behavior. How CDC42 is activated upon EFA6B depletion remains to be elucidated. A direct molecular mechanism could be through an ARH-GAP that would link Arf6 activation to CDC42 inhibition[59], alternatively, the enrollment of the CDC42-GEFs to

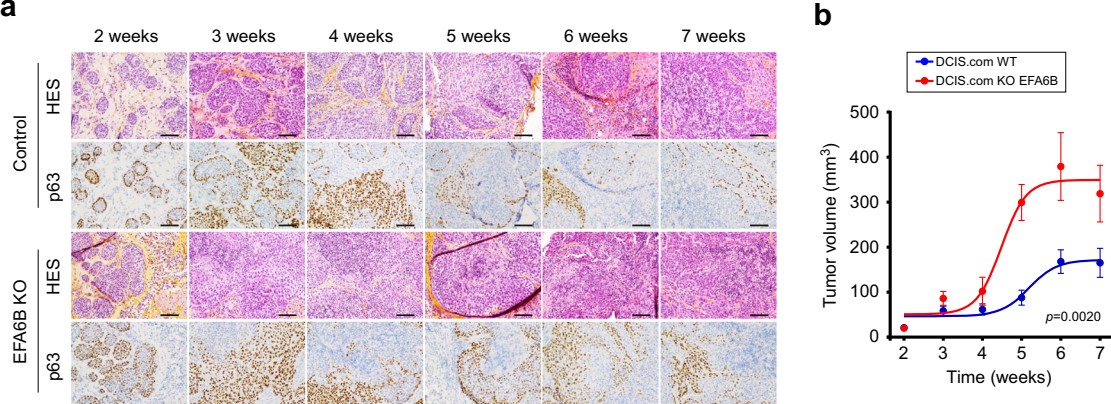

**Fig. 8 EFA6B KO in DCIS.com cells stimulates tumor growth and invasion. a** Representative images of HES (hematoxylin–eosin–saffron) coloration and p63 immunohistochemistry staining of the indicated tumor xenograft at the indicated time. Scale bar, 100 μm. **b** Tumor xenograft curve average ± SEM of WT (blue) and KO EFA6B (red) DCIS.com cells. $n = 4$ except at 4 weeks $n = 6$ for WT, $n = 5$ for KO 55. Two-way ANOVA $p$-value. Source data are provided as a Source Data file.

activate CDC42 could be the consequence of the EMT program, or in response to the remodeled ECM.

Heterozygous KO clones were as invasive as the homozygous ones, emphasizing the importance of the levels of expression of EAF6B in repressing the invasive potential of normal mammary cells. This observation is in agreement with many of our previous results indicating that EFA6 protein expression levels are a critical determinant of their physio-pathological functions[7,60–62]. Our results show that *PSD4*/EFA6B behaves as a haplo-insufficient gene with respect to the acquisition of an invasive phenotype. Since in BC patients, we found a correlation between the severity of the clinico-pathological data and the gradual loss of expression of *EFA6B* messenger[7], we propose that *PSD4*/EFA6B could act as a haplo-insufficient tumor-antagonist gene[63,64]. In BC, mutations and copy number alterations of *PSD4*/EFA6B gene are very rare[7,35]. It, therefore, seems more likely that the loss of EFA6B expression in BC patients is due to events affecting its messenger and/or protein abundance[7,62].

There have been no genomic studies showing *PSD4*/EFA6B to be a tumor suppressor gene, and our orthotopic xenograft experiments with MCF10A WT or EFA6BKO cells injected into immunosuppressed mice did not produce any tumor arguing that the loss of EFA6B alone is not a driver mutation. Yet, non-transforming secondary driver mutations provide selective advantages to transformed cells[65]. The discovery that the simple loss of EFA6B can model cancer invasion implies that such alteration can orchestrate many of the steps of the invasion cascade independently of an oncogenic mutation, congruent with the two-hit hypothesis, one that will transform the cell and another one that will help overcome the control by the microenvironment[66]. Thus, the loss of expression of EFA6B could act as a secondary driver mutation conferring invasive properties and high metastatic susceptibility to transformed cells. Consistent with this hypothesis, we have shown here that invasive BCs had a diminished expression of *PSD4* when compared to pre-invasive in situ BCs and that EFA6BKO cells have a transcriptomic signature that shares several gene ontologies with signatures of progression from DCIS to IDC. Furthermore, *PSD4* expression also decreased from the primary BC to the metastatic sample in a series of 29 matched pairs, and *PSD4* downregulation was associated with poor-prognosis features and shorter DFS in a large series of non-metastatic BC. These are key observations with respect to the treatment of patients diagnosed with DCIS and early invasive BC[67].

To establish the causal relationship between EFA6B KO and invasion in vivo, we used the DCIS.com xenografts model that mimics DCIS-to-IDC transition. We observed an enhanced percentage of p63-positive cells, faster acquisition of the IDC phenotype, and increased tumor growth in EFA6BKO xenografts. Interestingly, p63 expression has been associated with ECM-mediated signaling, EMT, collective invasion[68] and has been proposed together with MMP14 to contribute to DCIS invasion[69]. Together with our patients' studies, these results identify EFA6B as a critical regulator of invasion and progression of high-grade BC.

Further studies of this EFA6B-regulated pathway could help to better understand, predict and treat the progression of DCIS towards invasive BC and of invasive BC towards the lethal metastatic form of this disease.

## Methods

**Cells, antibodies, and reagents**. MCF10A and HEK-2923T cells were obtained from ATCC (LGC Standards, France) and grown in DMEM/F-12 (1:1), horse serum 5%, non-essential amino acids 1%, insulin 10 μg/ml, hydrocortisone 1 μg/ml, EGF 10 ng/ml, cholera toxin 100 ng/ml and penicillin (100 μg/ml)-streptomycin (100 μg/ml). HMLE cells were obtained from Dr. R.A. Weinberg (Whitehead Institute for Biomedical Research, Cambridge, MA, USA)[70] and grown in DMEM/F-12 (1:1), fetal calf serum 10%, insulin 10 μg/ml, hydrocortisone 0.5 μg/ml, and penicillin (100 μg/ml)-streptomycin (100 μg/ml). DCIS.com cells were obtained from Dr. P. Chavrier (Institut Curie, Paris, France) and grown in DMEM/F-12 (1:1), L-glutamine 2 mM, horse serum 5%. All culture reagents were from Invitrogen (Fisher Scientific, France) except for the fetal calf serum (Dutscher, France). Unless otherwise indicated, all other reagents were from (Sigma-Aldrich, France). All secondary antibodies and fluorescent probes were from Molecular Probes (Invitrogen). For the list of primary antibodies refer to Supplementary Table 3.

**3D culture**. 3D culture was performed using rat tail Collagen I (Corning®, Fisher Scientific, France). The collagen I solution was neutralized using 1 N NaOH and diluted in PBS to a final concentration of 1 mg/ml for the contractility assay, and 2 mg/ml for the invasion assay.

**CRISPR/Cas9 knock-out**. The *PSD4* KO mutation was obtained using the CRISPR/Cas9 gene-editing technology[71]. Two guided RNAs targeting the exon 1 of human *PSD4* were selected from the crispr.mit.edu/ and crispor.tefor.net/ websites. Off-target specificity was assessed by BLAST analysis of the human genome. The guided RNAs were cloned separately into the vector pSpCas9(BB)-2A-GFP (PX458) encoding for Cas9 and GFP (Addgene plasmid #48138). Sequences were as follows: PSD4-Z1T8-pX458-FOR1 forward 5′ CACCGaggatccaccggagcctttcg 3′, PSD4-Z1T8-pX458-REV1 reverse 5′ AAACcgaaaggctccggtggatcctc 3′, PSD4-Z2T36-pX458-FOR2 forward 5′ CACCGgttctctgagcaaggactcgcc 3′, PSD4-Z2T36-pX458-REV2 reverse 5′ AAACggcgagtccttgctcagagaac 3′. The control SgRNA (pLKO1-puro U6, #50927) was from Addgene. HMLE and MCF10A cells were transfected using Lipofectamine 3000 (Invitrogen) according to the manufacturer's

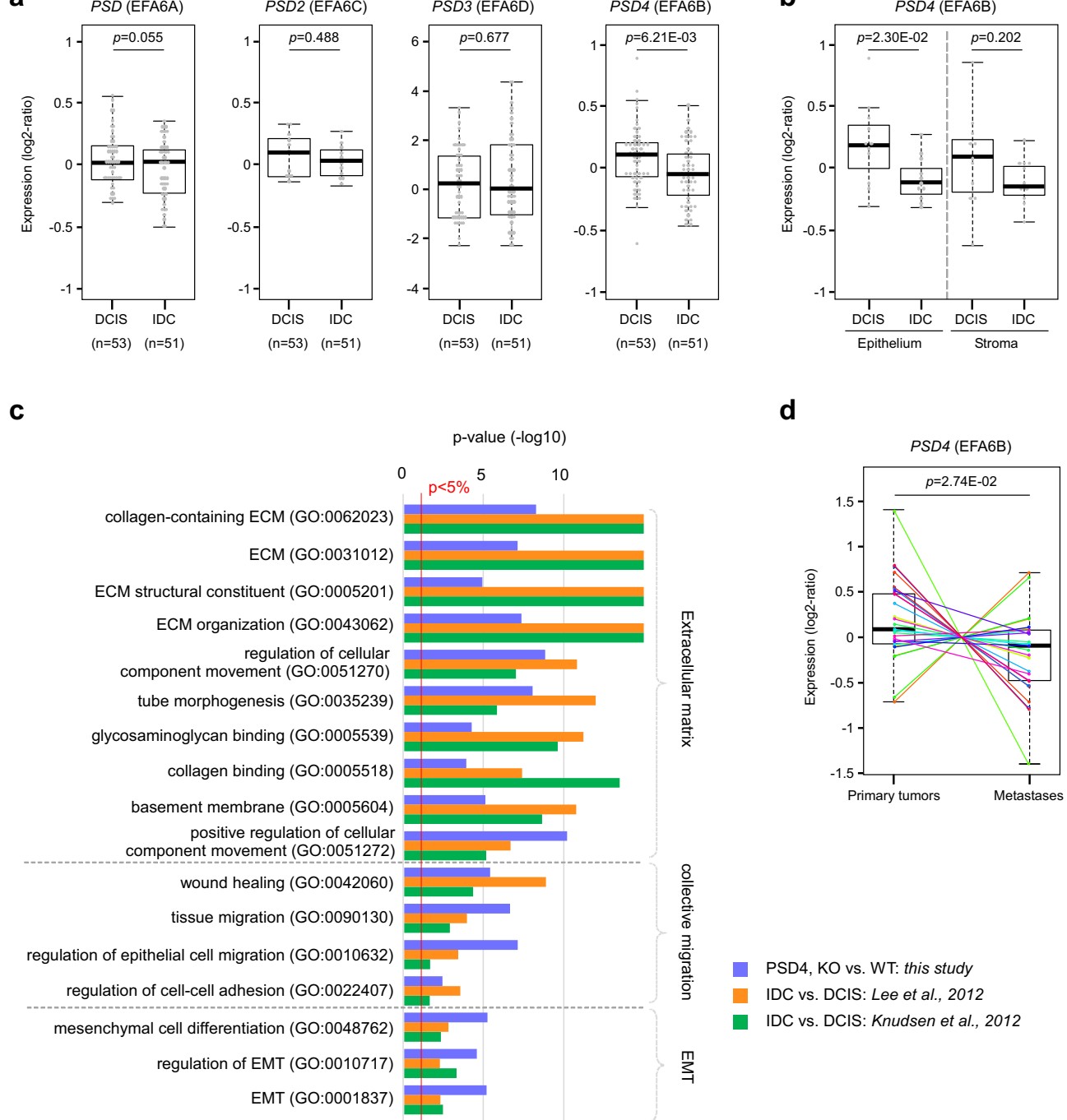

**Fig. 9 EFA6B/PSD4 expression is downregulated in human clinical BC samples endowed with invasive properties. a** Expression of the four EFA6 gene isoforms in DCIS (ductal carcinoma in situ) compared to IDC (invasive ductal carcinoma). The *p*-values are for the Student's *t*-test. **b** Expression of *PSD4* is selectively decreased in the epithelial compartment of DCIS ($n = 11$). The *p*-values are for the Student's *t*-test. **c** Ontologies associated with *PSD4* knock-out in MCF10A cells (blue), and with the comparison of expression profiles of IDC versus DCIS samples in the Lee's (orange) and Knudsen's (green) data sets[33,34]. **d** Expression of *PSD4* is decreased in the metastatic samples versus the paired primary BC samples ($n = 29$). The corresponding primary BC−metastasis pairs are connected by thin colored lines. The *p*-value is for the paired Mann–Whitney test. Source data are provided as a Source Data file.

recommendations. 24 h post-transfection, GFP positive cells were sorted and cloned in 96-well culture plates using the BD FACSAria III (BD Biosciences). The isolated clones were screened by immunoblot and the mutations validated using the Surveyor® assay (Integrated DNA Technologies) followed by genomic sequencing of the targeted sequence. The three different populations of HMLE were sorted according to their expression of EpCAM and CD49f as shown in Fig. 2 before next-day transfection. The DCIS.com KO EFA6B cell population was obtained by simultaneous infection with the aforementioned lentiviral vector PSD4-Z1T8-pX458 and a 1/100 MOI dilution of the lentiviral vector

pX459-puromycin plasmid (Addgene #62988) followed by selection with puromycin 1μg/ml.

**RNA isolation and RT-qPCR**. Total RNA was isolated using the Tri Reagent (Sigma) and treated with Ambion™ Dnase I (Invitrogen) following the manufacturers' instructions. RNA quality was tested using an Agilent Bioanalyzer. 2μg of total RNA was denatured at 65 °C for 10 min and incubated for 1 h at 50 °C in the presence of 2.5 mM dNTP, 100U Superscript III (Invitrogen) using 0.5μg oligo(dT)

15 primer in a total volume of 20 μl, followed by inactivation at 70 °C. A control PCR reaction of the reverse transcription was performed with human GAPDH forward (gaacatcatccctgcatcc) and reverse (ccagtgagcttcccgttca) primers with Q5 High fidelity DNA polymerase according to the manufacturer's instructions (New England BioLabs®). Real-time PCR was carried out with The LightCycler® 480 SYBR Green I Master (Roche Life Science) in triplicates and analyzed using LightCycler® 480 Software, v1.5 (Roche). The expression of each gene was normalized to the HPRT1 or GAPDH housekeeping genes and relative levels were calculated on the basis of the comparative cycle threshold Ct method $(2-\Delta\Delta Ct)$ where ΔΔCt is the difference in Ct between target and reference gene. For the list and sequence of RT-qPCR oligonucleotides refer to Supplementary Table 4.

**Transcriptomic analyses.** DNA-microarrays were used to define and compare the transcriptional profiles of MCF10A harboring *PSD4* homozygous (i.e., KO55, $N = 3$) or heterozygous (i.e., Het2.9, $N = 2$) knock-out and WT MCF10A as control ($N = 2$). Experiments were done as recommended by the manufacturer (Affymetrix, Thermo Fisher) from 100 ng of total RNA for each sample using the Affymetrix GeneChip HuGene 2.0 ST arrays. Expression data were normalized by RMA with the non-parametric quantile algorithm in R using Bioconductor and associated packages (version 3.5.2; http://www.cran.r-project.org/). Before analysis, expression data were filtered to remove probes with low and poorly measured expression and standard deviation inferior to 0.25 log$_2$ units across samples, resulting in 5,640 genes. We compared the expression profiles between the MCF10A harboring *PSD4* knock-out ($N = 5$) and the MCF10A control ($N = 2$) using a moderated *t*-test with empirical Bayes statistic[72] included in the limma R packages (version 3.38.3). False discovery rate (FDR)[73] was applied to correct the multiple-testing hypothesis. Significantly differentially expressed genes were defined by the following thresholds: *p*-value < 0.05, *q*-value < 0.15 and fold change FC > |2x|. The same supervised analysis was applied to the IDC versus DCIS comparison using the clinical samples from two publicly available data sets, which included 51 IDC and 53 DCIS for the Lee's set[33] and 10 IDC and 10 DCIS (epithelial samples) for the Knudsen's set[34]. Ontology analysis of the resulting gene list was based on the Gene Ontology (GO) terms using the Database for Annotation, Visualization, and Integrated Discovery (DAVID; david.abcc.ncifcrf.gov/). Furthermore, to explore the impact of *PSD4* knock-out in MCF10A on the matrisome and its receptors, we applied Gene Set Enrichment Analysis (GSEA) (http://www.broadinstitute.org/gsea/) for comparing the expression profiles of *PSD4* KO *versus PSD4* WT MCF10A cells. GSEA was based on two gene sets defined by the GO terms list 'Cell–cell adhesion' and a gathering of the KEGG gene sets Focal Adhesion ('FA', hsa04510), ECM-receptor interaction ('ECM', hsa04512), and the human matrisome database (version August 2014; Hynes Lab; Naba A, Ding H, Whittaker CA, Hynes RO. http://matrisomeproject.mit.edu) defining our "Matrisome+receptors" gene set. We used the class differential metric for ranking these filtered genes, weighted enrichment statistic for computing enrichment score (ES) of each gene set tested, and 1000 permutations to evaluate significance as parameters for the GSEA.

To determine *PSD4* mRNA expression in clinical BC samples, we collected and analyzed publicly available transcriptomic data. First, we compared the expression of the four *PSD* genes in a series of 53 ductal carcinomas in situ (DCIS) and 51 invasive ductal carcinomas (IDC), including 11 cases for which separate samples of tumor epithelium and adjacent stroma had been generated by laser-capture microdissection (LCM) before profiling[33]. Expression profiling was based on DNA microarrays and a comparison of expression levels between DCIS and IDC was done using the Student's *t*-test. Second, we analyzed expression from 29 matched metastasis-primary BC pairs, collected through two data sets: TCGA[35] for 7 pairs, and Vareslija's study[36] for 22 pairs, all having been profiled using RNA-seq. The comparison of *PSD4* expression levels between metastases and primary BC was done using the paired Mann–Whitney test. Finally, in order to search for correlations with clinico-pathological parameters in invasive BC and to validate our previous results[7] in independent samples, we analyzed DNA microarray- and RNA-seq-based data generated across 4 retrospective data sets not included in our previous report[7] (Supplementary Table 1). This resulted in a total of 3613 non-redundant primary invasive BC and four normal breast (NB) samples. Analysis was done as previously reported[7]. PSD4 downregulation was defined by a ratio tumor/NB ≤ 0.5, and no down-deregulation by a ratio >0.5. To avoid biases related to immuno-histochemical and pathological analyses across different sets and to increase the amount of available data, ER and ERBB2 statutes were defined as positive or negative using respective mRNA expression data of *ESR1* and *ERBB2* as described previously[7]. To compare the distribution according to categorical variables, we used the correlation of *PSD4* expression (down versus no down) with the clinico-pathological features using the Student's *t*-test for continuous features and Fisher's exact test for discrete features. Disease-free survival (DFS) was calculated from the date of diagnosis until the date of relapse or death. Follow-up was measured from the date of diagnosis to the date of last news for patients without event. Survivals were calculated using the Kaplan-Meier method and curves were compared with the log-rank test.

**Transient transfection.** Transient transfection of siRNAs was performed using RNAiMAX (Invitrogen) according to the manufacturer's instructions. All siRNAs were from the Dharmacon ON-TARGETplus collection (Horizon Discovery). For the list and sequence of siRNA oligonucleotides refer to Supplementary Table 5.

**Stable transfection.** HEK-293T cells were transfected with the human *PSD4* containing pLV-Neo-CMV lentivirus (Vector Builder, IL, USA) together with the 3rd generation lentiviral packaging plasmid (Addgene): pMDLG/pRRE (Gag and Pol), pRSV-Rev (Rev), and pMD2.G (VSV-G envelope) using JETPEI (Polypus Transfection). Supernatants containing the lentiviruses were collected at 48 and 72 h post-transfection. MCF10A cells were transduced with the filtered supernatants in the presence of 10 μg/ml of Polybrene (Sigma-Aldrich). Transduced cells were selected with 250 μg/ml of G418 (Sigma).

**Immunoblot.** Cells grown on plastic dishes washed with PBS were lysed with an SDS lysis buffer (0.5% SDS, 150 mM NaCl, 5 mM EDTA, 20 mM Triethanolamine–HCl pH 8.1, 1 mM PMSF). The lysate was heated at 95 °C for 10 min and then thoroughly vortexed for 15 min. After centrifugation at $16,000 \times g$ for 20 min at room temperature, the supernatant was transferred into a new tube containing 5× Laemmli buffer and further boiled for 5 min. For protein analysis, cell lysates were loaded into SDS–PAGE and transferred onto a nitrocellulose membrane. Membrane blocking and secondary antibodies dilutions were done in PBS 5% non-fat dry milk, primary antibodies were diluted in PBS 3% BSA. The proteins were revealed by chemiluminescence (ECL™, Amersham France) using secondary antibodies directly coupled to HRP. The membranes were analyzed with the luminescent image analyzer Fusion (VILBER, France) and band intensity quantified using the image analyzer software AIDA (Elysia-raytest, Germany). Results were normalized to the loading controls and to the control sample arbitrarily set at 1. Uncropped and unprocessed scans of all blots are presented in the Source Data file.

**Flow cytometry.** Cells were detached using Accutase (Stemcell technologies, NC, USA) and washed 3 times in PFE (PBS, 2 mM EDTA, 2% Fetal Bovine Serum). The cells were incubated at 4 °C for 30 min in the presence of the indicated antibodies diluted in PFE. After washes, cells resuspended in cold PFE were examined by BD LSRII Fortessa™ cell analyzer. The ALDEFLUOR kit (Stemcell Technologies) was used to quantify the ALDH enzymatic activity. The gating strategy is provided in Supplementary Fig. 8. Results were processed using Kaluza v2.1 or FlowJo v10 software.

**Invasion assay.** Cells transfected with siRNA were plated in collagen 24 h after transfection. Human mammary individual cells (100 cells/μl) re-suspended in 200 μl of collagen (2 mg/ml) were plated in chambered cover glass (Lab-Tek, Fisher Scientific), and incubated for 30 min at 37 °C in 5% CO2 before the addition of complete medium. After 48 h, protrusions were quantified using a phase-contrast microscope (Leica) by counting 100 cellular aggregates per well. Cells with at least one membrane extension of at least 2 microns' length were considered invasive.

**Time-lapse imaging.** To form spheroids, $50 \times 10^3$ cells were incubated in complete medium for 48 h in poly-HEMA coated 35 mm dishes. Size homogenous spheroïds isolated by differential low-speed centrifugation were resuspended in 250 μl of fibrillary collagen I (2.5 mg/ml) and plated in 48-well plates. 24–48 h later images were acquired using a Cytation™ 5 imaging multi-mode reader (BioTek Instruments) in a controlled environment. Time-lapse images were captured every 2–4 h and up to 64–120 h with a ×4/0.13 or a ×20/0.45 phase contrast objective using the Gen5 software (Biotek Instruments).

**Fluorescent gelatin degradation assay.** Oregon Green 488-labeled gelatin was from Molecular Probes (Invitrogen). Sterilized coverslips (18-mm diameter) were coated with 50 μg/ml poly-L-lysine for 20 min at RT, washed with PBS, and fixed with 0.5% glutaraldehyde for 10 min. After 3 washes with PBS coverslips were inverted on a 40 μl drop of 0.2% fluorescently labeled gelatin in 2% sucrose in PBS and incubated for 10 min at RT. After washing with PBS, coverslips were incubated in a 5 mg/ml solution of borohydride for 3 min, washed three times in PBS, and incubated with 1 ml of complete medium for 30 min. $5 \times 10^4$ cells per 12-well were plated on the fluorescent gelatin-coated coverslips and incubated at 37 °C for 48 h. Cells were then washed three times with PBS and fixed with 4% PFA for 20 min and processed for labeling with Texas Red-Phalloidin and DAPI. The coverslips were mounted with Mowiol mounting medium. Cells were imaged on a confocal microscope (Leica TCS-SP5). For the quantification of the gelatin degradation, the total area of degraded matrix measured with ImageJ software was divided by the total area of each phalloidin-labeled cell as described elsewhere[74]. 90 cells were analyzed in three independent experiments.

**Quantification of cellular collagenolysis.** Overall, $2 \times 10^4$ MCF10A cells were re-suspended in 0.2 ml of 2 mg/ml collagen I solution loaded on a Lab-Tek 8-well chambered cover glass. After gelling for 1 h at 37 °C, a complete DMEM-F12 medium was added, and collagen-embedded cells were incubated 48 h at 37 °C in 1% CO2. After fixation in 4% PFA for 20 min, samples were incubated with collagen type I cleavage site antibody (Col1-2/3C) diluted 1:200 in PBS (5 μg/ml, 24 h at 4 °C), washed extensively with PBS, and counterstained with Cy5-conjugated anti-rabbit IgG antibodies, DAPI and Texas Red-phalloidin. Images acquisition was performed with a confocal microscope (Leica TCS-SP5) with a ×63 oil

objective. Quantification of degradation spots was performed with a homemade ImageJ program as described elsewhere[74]. The degradation index is the number of degradation spots divided by the number of cells in each cluster present in the field 6 aggregates of 30 cells were analyzed in 2 independent experiments.

**Immunofluorescence**. Cells were fixed in 4% paraformaldehyde for 20 min, washed three times in PBS, and permeabilized in PBS containing 0.05% saponin, 0.5% horse serum for 30 min at 37 °C. Then the cells were incubated with primary antibodies over-night and counterstained with appropriate fluorescent secondary antibodies, DAPI, or phalloidin. Images acquisition was performed with a confocal microscope.

**Co-localization quantification**. Cells were imaged by confocal microscopy on their bottom focal-plane to visualize their ventral invadopodia and minimize irrelevant signals from vesicular structures. All images of the same dataset were captured on the same day using the same acquisition settings. Co-localization of the indicated markers was then quantified by the ImageJ plug-in JACoP2 using Pearson's correlation analysis[75]. Briefly, a ROI of the inner cell area excluding the peripheral staining was drawn. Then, the thresholds of each marker were set and used for all the images of the same set. The MMP14-mCherry threshold was set low so as to only include large intense invadopodia structures. The Costes randomization test was run on the same ROI to verify the statistical significance of the co-localization.

**Confocal reflectance microscopy**. MCF10A spheroids were embedded in collagen I for 48 h at 37 °C, then spheroids were fixed, and immunofluorescence was performed as indicated above. The imaging of the orientation of the collagen matrix fibers was performed by confocal reflectance microscopy using a scanning confocal microscope (Leica TCS-SP5). The collagen gels were excited with a 488 nm laser, and a signal between 485 and 495 nm was collected.

**Contractility assay**. 70 µl of collagen mixed with human mammary cells (700 cells/µl) were plated in 96-well plates and incubated for 30 min at 37 °C in 5% $CO_2$ before the addition of a complete medium. The collagen gels were detached from the plastic using a pipet cone. Collagen area was quantified using ImageJ software.

**Pull-down assay**. MCF10A cells grown on collagen-coated plastic were lysed at 4 °C in 0.5% Nonidet P-40, 20 mM Hepes pH 7.4, 125 mM NaCl, 1 mM phenylmethylsulphonyl fluoride (PMSF), and a cocktail of protease inhibitors (Complete, Roche Diagnostics). The cleared lysates were incubated 4 h at 4 °C with 1.5 µM of the indicated GST-fused proteins and 30 µl of glutathione-sepharose CL-4B beads (GE Healthcare). After three washes in lysis buffer, the beads were boiled in Laemmli buffer, submitted to SDS–PAGE, and the indicated proteins were revealed by immunoblot. GST fused to the ARF6GTP-binding domain of ARHGAP10, the CDC42/RAC binding domain of PAK, or the RHO binding domain of Rhotekin were used to pull-down ARF6GTP, CDC42GTP/RAC1GTP, RHOAGTP, or RHOCGTP, respectively. Each pull-down was repeated at least three times. The quantification of the amount of proteins pulled-down was normalized to the input and to the control sample arbitrarily set at 1.

**Integrin antibody blocking assay**. MCF10A cells were trypsinized, washed with PBS, and re-suspended in a complete DMEM-F12 medium. The cells were then incubated for 30 min at 37 °C to allow for recovery of cell surface receptors. $5 \times 10^4$ cells were re-suspended in 0.2 µl of a 2 mg/ml collagen I solution and control or blocking antibodies were added. The cell suspension was loaded on an 8-well Lab-Tek chambered coverglass and left to jellify for 1 h at 37 °C. Complete DMEM-F12 medium containing the control or blocking antibodies was added on top of collagen-embedded cells. The samples were incubated 48 h at 37 °C in 1% $CO_2$. The development of protrusions was quantified as described above by counting 100 cell clusters, in 3 independent experiments.

**Xenograft experiments**. $10^5$ DCIS.com (WT or KO EFA6B) cells were prepared in PBS supplemented with Matrigel (v/v) (Corning, Bedford, MA) and injected subcutaneously into 8–9-week-old female nude mice. The tumor growth was monitored before resection with a digital caliper and the tumor volume calculated as follows: length × width$^2$ × π/6. Xenografts resected at different time points were snap-frozen in dry ice and stored at −80 °C for subsequent formalin fixation and staining. All experiments were done in agreement with the French Guidelines for animal handling and approved by the local ethics committee (Agreement n° 2016091517253478) of our local ethics committee Comité Institutionnel d'Ethique Pour l'Animal de Laboratoire (CIEPAL Azur).

**Ethics approval**. Public data come from published studies in which the patients consent to participate; the ethics and institutional review board were already obtained by the authors. Our study was approved by our institutional review board (Comité d'Orientation Stratégique, COS).

**Immunohistochemistry**. Paraffin-embedded formalin-fixed tumor blocks were cut and immunohistochemistry was performed on serial 4-µm deparaffinized sections. Immunohistochemistry was performed on a Dako Autostainer Link 48 biomarker platform. Staining was carried out with the anti-P63 antibody (clone 4A4-Diagomics #BSB3606) after antigen retrieval at pH 9 for 20 min at room temperature at a final dilution of 1/50. Staining was performed using Labelled Polymer SM802 FLEX/HRP with DAB substrate (Dako). Slides were counterstained with hematoxylin II (Dako). On a consecutive section, hematoxylin–eosin–saffron (HES) staining was performed on the Dako Coverstainer as recommended by the manufacturer.

**Statistics and reproducibility**. Unless otherwise indicated, all experiments were repeated independently at least three times with similar results, the number of repeats is indicated in the legend as (N) and the number of cells or cell aggregates analyzed as (n), error bars represent ± standard error (SEM) or standard deviation (SD) of the mean as indicated in the figure legends. Unless otherwise indicated in the legend, statistical significance was determined using the two-tailed Student's t-test or the one-way ANOVA test with Dunnett's multiple comparison, in which p-values < 0.05 were considered statistically significant. All statistical analyses have been performed with GraphPad Prism software.

**Reporting summary**. Further information on research design is available in the Nature Research Reporting Summary linked to this article.

## Data availability

All the data supporting the findings of this study are available upon request to the corresponding author. Generated microarray data were deposited on Array-Express repository (E-MTAB-10063, https://www.ebi.ac.uk/arrayexpress/experiments/E-MTAB-10063). For public data sets of invasive breast cancer, all data analyzed were downloaded from the Gene Expression Omnibus (GEO, https://www.ncbi.nlm.nih.gov/geo/), Genomic Data Commons (GDC, https://portal.gdc.cancer.gov/), and European Genome-phenome Archive (EGA, https://ega-archive.org/) databases as detailed in Supplementary Table 1. DCIS public data sets were downloaded from GEO (GSE41228 and GSE33692) for Lee's and Knudsen's data sets, respectively[33,34]. Source data are provided with this paper.

## Material availability

All materials used in this study are readily available from the authors.

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

## Acknowledgements

This work was supported by the Centre National de la Recherche Scientifique (CNRS) and the Fondation ARC pour la Recherche sur le Cancer. RF was supported by the

French Government (National Research Agency, ANR) through the "Investments for the Future" programs LABEX SIGNALIFE ANR-11-LABX-0028 and IDEX UCAJedi ANR-15-IDEX-01. MVR was supported by a CONACYT fellowship from the Mexican government. PF, ML, DB and FB are supported by the Ligue Nationale Contre le Cancer (label) and Fondation Groupe EDF. The authors acknowledge the flow cytometry and microscopy facility from the « Institut de Pharmacologie Moléculaire et Cellulaire » part of the « Microscopie Imagerie Côte d'Azur » GIS IBiSA labeled platform. We would like to thank Pr. R.A. Weinberg (Whitehead Institute for Biomedical Research, MA, USA), for providing the HMLE cells, and Dr. P. Chavrier (Institut Curie, Paris, France) for sharing reagents and his expertise.

## Author contributions

F.L. supervised the study. F.L. and M.F. conceptualize the study. F.L., R.F., M.V.R., F.B., and M.L. wrote the manuscript. F.B. and P.F. performed, collected, and analyzed the cell gene expression profiling and patients' data sets. M.L., O.C., and A.F. performed and analyzed the xenografts and immunohistochemistry experiments. S.A. performed the video-microscopy and provided support for imaging analysis. F.L., M.F., R.F., M.V.R., M.P., S.D., and V.V. performed and analyzed all the other experiments. D.B. provided constructive comments and discussion.

## Competing interests

The authors declare no competing interests.
