## [Peer Review File · Nature Communications]

Reviewers' comments:

Reviewer #1 (Remarks to the Author):

The in situ to invasive transition is a pivotal early step in the metastatic process whose underlying molecular mechanisms remain poorly understood. Here, Racha and colleagues investigate the contribution of EFA6B to invasion of normal human mammary epithelial cells into collagen-gels. Using CRISPR, they show that EFA6B KO have increased collective invasion, increased MMP14 protease activity, and increased ITGB1-dependent invadopodia. These changes appear associated with transcriptional changes consistent with an EMT, and the authors provide supportive evidence that the invasion phenotype at least appears dependent on CDC42 and its effector pathways MRCK-pMLC and N-WASP-ARP2/3. Supporting the human disease relevance of their findings, they show gene expression data indicating reduced PSD4 (EFA6B) in IDC compared with DCIS, and worse disease free survival of PSD4 low tumors in published breast cancer patient datasets. Overall, the findings in this study are technically sound- with a few important areas of confounding outlined below. When taken together with their prior JCS and Cancer Research studies, it is convincing that EFA6B has strong effects on epithelial organization and migratory activity in MCF10A mammary cells. However, an important shortcoming is that experiments are lacking which establish the in vivo relevance of EFA6B (e.g. with xenograft models). Thus at this time, their core claims that EFA6B controls the invasive potential of mammary cells, and that this could be an important early step in progression to invasive human cancer is not sufficiently substantiated. Given their prior published work in EFA6B and metastatic breast cancer, the study in its current form might be better suited for a more specialist cell biology journal.

Major Comments:

- 1) The bedrock of this paper is the use of CRISPR to knockout EFA6B. As outlined in the methods, Racha and colleagues start with MCF10A WT cells, and generate KO lines by single cell cloning. An important confounding is that passaging MCF10As in this way might be a selective pressure in itself, that could select for a subpopulation that is inherently more happy as single cells, and has undergone an EMT. The authors should generate CRISPR ko-lines with a control-guide and determine their invasive behavior to rule out this possibility.
- 2) The authors claim that EFA6B unleashes collective invasion but it is not clear how collective the cells really are in Figure 1c. Mesenchymal cells can collide with each-other dynamically, rather than traveling as a cohesive cohort of cells. The authors should perform time-lapse imaging starting with similar sized aggregates, and more precisely determine single cell vs. collective invasion. Based on KO 55 it may be more accurate to say there is increased invasion of both types. Further, the authors show invasion for KO55 but not KO50 or KO2 in Fig 1C- do these also show collective invasion phenotypes? It would be important to show since KO50 and KO2 have even stronger EMT protein expression patterns (Fig 5a).
- 3) Currently the study lacks proof of in-vivo relevance. The authors state in the discussion that "our orthotopic xenograft experiments of MCF10A EFA6B KO cells in immunosuppressed mice did not produce any tumor arguing that the loss of EFA6B alone is not a driver mutation." Is it the case that EFA6B KOs did not form tumors but EFA6B WT did? Or that MCF10As did not form tumors at all? It is not clear from the wording and without the data. In the absence of in vivo support, the findings in this study are not generalizable. It is conceivable that the ex-vivo invasion observed here is not directly correlated to invasion in vivo tumors. If MCF10As are not a good model, a model that is transplantable like DCIS.COM (see Kornelia Polyak's myoepithelial barrier paper Cancer Cell 2008) could be helpful to test if EFA6B KO DCIS.COM transplants are more prone to forming invasive cancers.
- 4) Racha et al. provide good evidence for downstream signal transduction pathways activated in EFA6B KOs, and this is perhaps the most novel piece of this study when put into context of their prior work. However, a key step missing is how EFA6B induces an EMT. Is this an irreversible EMT? The authors should consider effect of EFA6B rescue (perhaps inducible rescue) and test its effects on EMT reversal. This could become a very neat tractable model to tease apart how EFA6B is able to remodel and reverse the EMT phenotype.

5) The magnitudes of effect in Figure 8 are small, and the authors should qualify their conclusions about the limitations of bulk RNA-seq in this context. It seems unlikely that all the DCIS or IDC cells will be homogenous. The log₂ ratio median difference in Figure 8a and 8b look to be less than 2 fold. Why do the authors think given their haploinsufficiency model that such small changes in EFA6B could strongly impact invasion? Do the authors see evidence of EMT signatures induced in IDC versus DCIS, or ECM signature genes as outlined in Figure 8c?

Minor comments:

1) Figure 8E has similarities to their paper Zangari et al. Cancer Research 2014, Figure 5E- how is this different conceptually?

2) With the luminal/basal experiments (figure 2) do cells maintain luminal phenotype in collagen or acquire basal myoepithelial characteristics?

3) Is there evidence EFA6B directly interacts with cdc42? Do they have any proposed mechanisms for how this might work?

4) The authors describe in the text a more or less complete EMT, but KO 55 shows persistent E-cad expression. Is this more hybrid or partial EMT? Does Ecad localize to intercellular contacts in their model in EFA6B KOs- supporting a collective invasion phenotype. If not E-cad what is holding cells together, N-cad?

5) At several points the author says "invalidated" whereas it seems to mean "knocked out"

6) The authors should provide more detail on rationale for MCF10A model which normally fails to form tight junctions in standard tissue conditions- if EFA6B is a tight junction regulator, why is MCF10A a good model to study its impact?

7) In the legends they report N=3 average SEM, but they do not report the number of aggregates counted per condition per replicate.

8) given their prior paper showing association of EFA6B with claudin-low and aggressive TNBC, it might be interesting to focus Fig 8 on specifically TNBC IDC cases, in their comparisons.

Reviewer #2 (Remarks to the Author):

This is an interesting manuscript describing how EFA6B suppresses an invasive phenotype in breast cancer. There are some important mechanistic observations: inhibiting this molecule leads to transcriptional reprogramming to support invasion. Collective invasion is controlled via regulation of ECM interactions and via CDC42-MRCK-Myosin activation. Patient data supports this mechanism.

Before publication, a few comments need to be addressed:

1. What happens if you inhibit other collagenases apart from MMP14- like MMP2/MMP13, etc?
2. Better measurement of myosin light chain activity is needed (blot in figure 6 not convincing). Quantification of the wbs is recommended (and statistical analysis), and authors should use immuno-fluorescence to measure phospho-myosin and where is located in the invasive structures.
3. What happens after ko of RhoC? And its GTPase activity? This small GTPase plays a key role in metastasis. Or RhoA+RhoC combined depletion?
4. Quantification of pull downs needs to be provided, how many times where these pull downs performed? Statistical analysis is needed.
5. In general: in the paper there are no measurements of actual invasion just protrusive aggregates?

"After 48h, protrusions were quantified using a phase contrast microscope by counting 100 cellular aggregates per well. Cells with at least one membrane extension of at least 2 microns' length were considered invasive".

This is not an invasion assay per se. This just quantifies protrusions, which could be inefficient protrusions and could not lead to actual invasion. Authors need to measure cell/group of cells displacement over time (the whole cell needs to invade not just make a protrusion).

6. Figure 6A, Rho and Rac ko give an intermediate phenotype regarding contractility, not as strong as Cdc42, but there is an effect. Please discuss. Can authors show the phenotype in all assays after Rac and RhoA/C ko? Including cell morphology, invadopodia and cell invasion?

Reviewer #3 (Remarks to the Author):

EFA6A, B, C and D are guanine nucleotide exchange factors that activate the small GTPase ARF6. One of these, EFA6B, has been shown to promote cell-cell contact in epithelial cells, and to restore epithelial morphology to moderately transformed cells. Here the authors present evidence that loss of the EFA6B gene (PSD4) in a weakly transformed breast epithelial cell line, MCF10A, leads to epithelial/mesenchymal transition (EMT) and collective invasion of the cells into a 3D collagen matrix. The data indicate that loss of EFA6B triggers transcriptional upregulation of EMT-inducing genes including SNAIL1, TWIST1 and ZEB1, a change in integrin repertoire and a downregulation of epithelial-specific genes encoding tight junction and cell-cell adhesion proteins. This transition was accompanied by increased activation of Cdc42 and formation of MMP14-enriched invadopodia, which are necessary for penetration of the collagen matrix. Importantly, patient-derived data indicate that EFA6B expression is reduced in invasive breast carcinoma, and that invasive tumors display transcriptional signatures similar to those of EFA6B-deficient MCF10A cells.

Overall, the study is well conceived and carefully conducted. In general the data are convincing and support the conclusions drawn by the authors. The inclusion of clinical samples and correlation with outcomes is important and strengthens the impact of the study. There are, however, several issues that need to be addressed:

1. All of the imaging data needs to be quantified. The authors' claims of colocalization appear over-interpreted in some cases, and would require unbiased quantitative analysis in any case. This is particularly notable in Fig. 4E, where the authors claim that β 1-integrin colocalizes with MMP14 in EFA6B-deficient cells, but not control MCF10A. This is clearly not the case, based on the images shown.

2. Similarly, none of the immunoblots are quantified. All blots should be representative of at least 3 independent experiments and quantified to demonstrate statistical significance. This is particularly important for the blots showing activation of Cdc42 but not Rac1 or RhoA, but should be done for all blots.

3. A related point – it is somewhat surprising that Cdc42 is the only Rho family GTPase activated under these conditions. Is this a property of 3D culture in collagen, or does it also occur in 2D if cells are plated on collagen-coated plastic?

4. On p4, the authors state that the transition from E-cadherin to N-cadherin, which occurs during EMT, was confirmed by RT-qPCR, but in the next paragraph state that they “did not notice a change in E- or N-cadherin expression”. What does this mean? That there was a transcriptional change without a corresponding change in protein expression?

5. In referring to Fig. 1D, the authors state that EFA6B-deficient cells exhibited more protrusions than controls, but that this was not accompanied by “migratory properties”. What does this mean? Isn't collective invasion considered migration?

6. Presumably many of the downstream effects of EFA6B loss are due to reduced activation of ARF6 at specific cellular locations. How do the authors think this leads to EMT, activation of Cdc42 and formation of invadopodia?

Valbonne, le 17 décembre 2020

Our detailed, point-by-point response to the referees follows:

Reviewer #1 (Remarks to the Author)

The in situ to invasive transition is a pivotal early step in the metastatic process whose underlying molecular mechanisms remain poorly understood. Here, Racha and colleagues investigate the contribution of EFA6B to invasion of normal human mammary epithelial cells into collagen-gels. Using CRISPR, they show that EFA6B KO have increased collective invasion, increased MMP14 protease activity, and increased ITGB1-dependent invadopodia. These changes appear associated with transcriptional changes consistent with an EMT, and the authors provide supportive evidence that the invasion phenotype at least appears dependent on CDC42 and its effector pathways MRCK-pMLC and N-WASP-ARP2/3. Supporting the human disease relevance of their findings, they show gene expression data indicating reduced PSD4 (EFA6B) in IDC compared with DCIS, and worse disease free survival of PSD4 low tumors in published breast cancer patient datasets. Overall, the findings in this study are technically sound- with a few important areas of confounding outlined below. When taken together with their prior JCS and Cancer Research studies, it is convincing that EFA6B has strong effects on epithelial organization and migratory activity in MCF10A mammary cells. However, an important shortcoming is that experiments are lacking which establish the in vivo relevance of EFA6B (e.g. with xenograft models). Thus at this time, their core claims that EFA6B controls the invasive potential of mammary cells, and that this could be an important early step in progression to invasive human cancer is not sufficiently substantiated. Given their prior published work in EFA6B and metastatic breast cancer, the study in its current form might be better suited for a more specialist cell biology journal.

Major Comments:

1) *The bedrock of this paper is the use of CRISPR to knockout EFA6B. As outlined in the methods, Racha and colleagues start with MCF10A WT cells, and generate KO lines by single cell cloning. An important confounding is that passaging MCF10As in this way might be a selective pressure in itself, that could select for a subpopulation that is inherently more happy as single cells, and has undergone an EMT. The authors should generate CRISPR ko-lines with a control-guide and determine their invasive behavior to rule out this possibility.*

We would certainly agree and we must apologize for not explaining better our procedure. In fact, similarly to the HMLE population, in the process of selecting EFA6B knock-out clones from MCF10A cells we also saved WT clones that were morphologically undistinguishable from the mother cell line. As requested by the Reviewer, we have also prepared by lentiviral infection a non-clonal MCF10A cell population expressing a sgRNA negative control (Addgene, plasmid #50927). As shown in the **Supplemental Fig.1a**, in contrast to the EFA6B KO55, when grown in collagen matrix all control cell lines and clones formed regular round aggregates with no sign of

invasive behavior. The reason we had shown the control WT clones solely for the HMLE cell line was because we had sorted the HMLE cells in three sub-populations and therefore it was critical in that case to demonstrate the non-clonal effect of *PSD4* knock-out. Our new data (**Supplementary Fig.1a**) further support the conclusion that the invasive phenotype observed in the EFA6B KO clones is not the result of a clonal selection process but a direct consequence of the loss of EFA6B expression.

2) The authors claim that EFA6B unleashes collective invasion but it is not clear how collective the cells really are in Figure 1c. Mesenchymal cells can collide with each-other dynamically, rather than traveling as a cohesive cohort of cells. The authors should perform time-lapse imaging starting with similar sized aggregates, and more precisely determine single cell vs. collective invasion. Based on KO 55 it may be more accurate to say there is increased invasion of both types. Further, the authors show invasion for KO55 but not KO50 or KO2 in Fig 1C- do these also show collective invasion phenotypes? It would be important to show since KO50 and KO2 have even stronger EMT protein expression patterns (Fig 5a).

Our conclusion that the invasion is collective came from the near absence of any single cells going far away from the spheroids in the thousands of images of invading aggregates at various time points we analyzed being from MCF10A or HMLE EFA6B KO clones. Thus, there was no evidence supporting the single cell invasion mode. We have added in **Fig.1c** representative images of the clones KO2 and KO50 that demonstrate the same invasive phenotype regardless of their apparent more pronounced EMT signature. Certainly time-lapse imaging is more convincing to demonstrate the collective invasion. We are now presenting two movies for MCF10A WT, KO2 and KO55 in **Supplementary Movies 1-2** that support our initial conclusion. As previously observed by others, MCF10A WT (WT or MCF10A SgRNA negative control) aggregates placed in collagen extend and retract membrane protrusions while rotating on themselves, yet we rarely observed chains of cells extending out into the collagen matrix (**Supplementary Movie 1, 20X**). In striking contrast, EFA6B KO2 and KO55 aggregates rapidly invaded the matrix in a collective fashion.

Sometimes, single cells can be noticed breaking transiently apart but they always connected back to their aggregate or invading chain of origin (for examples, see top right movie KO55 20X and top of the movie to the right KO2 4X). Also we observed, that the membrane protrusions or their extremities brake away leaving material at distance that seem to prepare for cell invasion, as we noticed that in most cases invading cells move out in the direction of these protrusions remnants deposited formerly. Note that the WT aggregates appear smaller because they remain compact, as compared to the KO aggregates in which cells are more loosely attached and from which extending peripheral cells tend to move away. We are confident that these new pieces of evidence support our conclusion that the mode of invasion of EFA6B KO cells in the fibrillary collagen I matrix is collective.

3) Currently the study lacks proof of in-vivo relevance. The authors state in the discussion that “our orthotopic xenograft experiments of MCF10A EFA6B KO cells in immunosuppressed mice did not produce any tumor arguing that the loss of EFA6B alone is not a driver mutation.” Is it the case that EFA6B KOs did not form tumors but EFA6B WT did? Or that MCF10As did not form tumors at all? It is not clear from the wording and without the data. In the absence of in vivo support, the findings in this study are not

generalizable. It is conceivable that the ex-vivo invasion observed here is not directly correlated to invasion in vivo tumors. If MCF10As are not a good model, a model that is transplantable like DCIS.COM (see Kornelia Polyak's myoepithelial barrier paper Cancer Cell 2008) could be helpful to test if EFA6B KO DCIS.COM transplants are more prone to forming invasive cancers.

To clarify the wording, the sentence has been changed to: "our orthotopic xenograft experiments for MCF10A WT cells or MCF10A EFA6B KO cells injected into immunosuppressed mice did not produce any tumor...."

We have performed the experiment suggested by the reviewer. PSD4 was knocked-out in the DCIS.com cell line by lentiviral infection and puromycin selection without cloning to obtain an EFA6B KO cell population. The subcutaneous xenograft experiment in SCID mice was performed by following exactly the procedure described in Hu and colleagues¹. The volume of each tumor was recorded overtime and the tumor samples collected at different times post-injection were analyzed by HES and p63 staining to monitor the invasion. The results are presented in the new **Fig.8 and Supplementary Fig.7a**. DCIS.com tumors started to invade at week 3-4 with clear invasion at week 5 similarly to previous studies¹⁻⁵. In the EFA6B KO xenografts, invasion was detected at week 2 and was more pronounced at all times compared to control DCIS.com. Furthermore, EFA6B KO invasion and increase in the percentage of p63-positive cells preceded an accelerated tumor growth rate. These results indicate a causal relationship between the loss of EFA6B and a more efficient invasion. Together with our patients' tumor results, it strongly indicates that a decrease in EFA6B expression would confer pro-invasive capacities to human mammary tumors. We thank the reviewer for their encouragement.

4) Racha et al. provide good evidence for downstream signal transduction pathways activated in EFA6B KOs, and this is perhaps the most novel piece of this study when put into context of their prior work. However, a key step missing is how EFA6B induces an EMT. Is this an irreversible EMT? The authors should consider effect of EFA6B rescue (perhaps inducible rescue) and test its effects on EMT reversal. This could become a very neat tractable model to tease apart how EFA6B is able to remodel and reverse the EMT phenotype.

I would agree that the description of the signaling pathways activated upon EFA6B KO is novel. However, I would say that the most novel piece of information of our study is the discovery that the knock-out of EFA6B is sufficient to induce an **invasive** behavior *in vitro*, which was not expected based on our previous work and knowledge of the EFA6 family. Considering our poor understanding of the molecular mechanisms controlling early steps of the metastatic process, as stated by the Reviewer, it makes this discovery not only novel but of serious interest. This is even more true in light of our xenograft results.

To answer the second question first, the rescue experiment was already presented in **Fig.1** and showed that the KO55 cells re-expressing EFA6B behaved similarly to the non-invasive WT cell line when grown in 3D collagen matrix. We have obtained similar results with the clones KO2 and KO50 (not shown). From these experiments, we had concluded that the effect of the sgRNA EFA6B was specific to the knocking-out of the *PSD4* gene.

Nevertheless, we had not studied in detail the EMT signature. In this current version, the western-

blot analysis of the classical EMT markers is now shown in the **Supplementary Fig.3f**. Re-expression of EFA6B was performed using lentiviral infection at various MOI. Interestingly, upon re-expression of normal levels of EFA6B we observed a recovery of the epithelial markers E-cadherin and Cld1. In contrast, the mesenchymal markers N-cadherin and vimentin did not decrease, N-cadherin levels even increased upon re-expression of EFA6B. No additional effects were observed upon overexpression up to 2.7 fold. The transcriptomic analysis of EFA6B re-expressing cells confirmed that the cells did not display the normal epithelial signature. Our finding underlies the complexity of defining epithelia-mesenchymal status and suggests that reversal of invasive behavior can be obtained by driving the re-expression of the epithelial markers without decreasing the levels of the mesenchymal markers.

It also indicates that the levels of expression of EFA6B control epithelial markers levels, which brings us back to the first question: how does EFA6B regulate EMT? It is an open question and considering our current knowledge, it will require an extensive amount of work to resolve it. Thus, at this point, we have no definitive answers but some hypotheses. One possibility is that the loss of EFA6B would cause the disassembly of the cell-cell junctions leading to the redistribution/release of molecules activating directly or indirectly the EMT program. For example, transcription factors or transcriptional co-activators located at the TJ or AJ could translocate to the nucleus to drive EMT⁶⁻¹⁰. Another consequence of the cell-cell junction disassembly would be the loss of the permeability barrier bringing together growth factor receptors and their ligands which could induce EMT¹¹⁻¹³. The remodeling of the extracellular matrix could also contribute EMT and invasion. The gene expression profiling revealed that the matrisome signature reflected by the change of the cell surface expression of the integrin molecules was the most affected. By changing their immediate environment EFA6B KO cells could receive in return EMT driving and pro-invasive signals^{14,15}. A sentence has been added in the paragraph "EMT" of the Discussion that sums up these hypotheses. In conclusion, our new results further support the role of EFA6B in controlling the EMT phenotype and indicate that EFA6B re-expression can selectively revert epithelial markers expression. As mentioned by the Reviewer it makes our cell model an excellent system to analyze EMT reversal.

5) The magnitudes of effect in Figure 8 are small, and the authors should qualify their conclusions about the limitations of bulk RNA-seq in this context. It seems unlikely that all the DCIS or IDC cells will be homogenous. The log2 ratio median difference in Figure 8a and 8b look to be less than 2 fold. Why do the authors think given their haploinsufficiency model that such small changes in EFA6B could strongly impact invasion?....

We agree that the magnitude of differences in gene expression are small. This has been added in the text by replacing "a significant reduction of PSD4 expression" by "a small but significant reduction of PSD4 expression". However, our haploinsufficiency model suggests that small changes in PSD4 expression can induce modification of the phenotype. Even though for convenience we compare patients presenting at least a two-fold decrease with the "neutral" patients, we find a full spectrum of PSD4 expression, and histoclinical correlations analyses showed that a continuum existed from downregulated to neutral to upregulated levels of EFA6B/PSD4¹⁶. Null mutation in PSD4 are not found in BC¹⁶, the decrease of EFA6B expression is thus due to finely tuned post-transcriptional and/or post-translational regulation, such as ubiquitinylation¹⁷. In our previous study (Zangari et al.), we had found inter- and intra-tumoral heterogeneity of EFA6B protein levels by IHC. Hence, we believe that the small decrease in the average of mRNA across tumor patients' is

consistent with individual or small clusters of cells with much reduced protein levels. These clusters of cells may act as leader cells to breach the basement membrane for the rest of the tumor mass to follow. In the past, using an inducible-expression system we had found that slight differences in EFA6A exogenous expression were sufficient to affect polarity and tight junction assembly in MDCK cells^{18,19}. In this context, we believe that small differences in expression may have a significant impact on the cell behavior.

.....Do the authors see evidence of EMT signatures induced in IDC versus DCIS, or ECM signature genes as outlined in Figure 8c?

We compared the gene ontologies associated with our 296-gene signature to those associated with the genes that we found differentially expressed between DCIS and IDC; using the clinical samples from two publicly available data sets, which included 51 IDC and 53 DCIS for the Lee's set²⁰ and 10 IDC and 10 DCIS (epithelial samples) for the Knudsen's set²¹. Interestingly, many ontologies related to ECM organization, collective migration and EMT were in common between the three signatures. The analysis (**Fig.9c**, Results, Methods) have been included in the revised version.

Minor comments:

1) Figure 8E has similarities to their paper Zangari et al. Cancer Research 2014, Figure 5E- how is this different conceptually?

Indeed, the series presented in the original version of the manuscript, contained redundancy with the previously published series. In our previous paper¹⁶, we had analysed *PSD4* mRNA expression in 5,252 primary breast cancer samples, collected from several data sets including ours, and 2,930 patients had DFS information available. In the original version of the manuscript, we had analysed *PSD4* mRNA expression in an extended pooled data set including 8,464 primary breast cancers, including the 5,252 previous cases, and the DFS analysis was done in 6,156 patients, including the 2,930 previous patients with DFS available.

To answer the concern of the Reviewer, we have redone the analysis of *PSD4* mRNA expression in clinical samples by excluding the redundant samples. That allowed us to test the robustness of our previous results in an independent series of 3,613 cases, including 3,353 cases with informative DFS. The new result that demonstrates the reproducibility of our previous observations is now presented in **Supplementary Fig.7e**.

The results have been modified in the revised version as follows. The sentences ““*Finally, to confirm and extend our previous results¹⁶ on a larger series, we searched for correlation between *PSD4* mRNA expression and the clinico-pathological features of our updated large publicly available series of 8,464 invasive primary BC (Supplementary Table 3). From this cohort, a total of 530 tumors showed a two-fold or greater down-regulation of *PSD4*, using normal breast tissue as the standard.*” have been replaced by the following ones: “*Finally, to confirm our previous results¹⁶ on a large and independent series, we searched for correlation between *PSD4* mRNA expression and the clinico-pathological features of a publicly available series of 3,613 invasive primary BC (Supplementary Table 3). From this cohort, a total of 306 tumors showed a two-fold or greater down-regulation of *PSD4*, using normal breast tissue as the standard.*”

Analysis of correlation of *PSD4* expression-based classes with the clinicopathological variables found similar results as our previous *Cancer Research* paper and the initial version of the present paper, there is a correlation of *PSD4* down-regulation with younger patients' age, higher pathological grade and tumor size, ductal type, higher frequency of TN subtype. The **Supplementary Table S4** has been revised. The following sentence "*PSD4* down-regulation was associated (Fisher's exact test; **Supplementary Table 4**) with higher pathological grade ($p < 0.001$) and tumor size ($p < 0.001$), ductal type ($p = 0.002$), higher frequency of ER-negative ($p < 0.001$) and ERBB2-negative ($p < 0.001$) statuses...." has been replaced by "*PSD4* down-regulation was associated (Fisher's exact test; **Supplementary Table 4**) with younger patients' age, higher pathological grade and tumor size, ductal type, and higher frequency of TN subtype ($p < 0.001$),..."

Survival analysis also showed similar results as our previous *Cancer Research* paper and the initial version of the present paper with correlation of *PSD4* down-regulation with shorter DFS. The original Fig.8e has been revised. The following sentence "Within the 6,156 non-stage IV patients with follow-up available, the 5-year DFS was 75% (95CI, 74-76%) for the whole population, and 65% (95CI, 59-70%) and 75% (95CI, 74- 77%) in cases of down-regulation and no down-regulation respectively ($p = 4.75E-04$, log-rank test; **Supplementary Fig.7e**)." has been replaced by "Within the 3,353 non-stage IV patients with follow-up available, the 5-year DFS was 82% (95CI, 80-83%) for the whole population, and 69% (95CI, 62-76%) and 83% (95CI, 80-84%) in cases of down-regulation and no down-regulation respectively ($p = 5.71E-04$, log-rank test; **Supplementary Fig.7e**)."

The corresponding modifications have been done in the Methods section within the "Transcriptomic analyses" subsection, and the **Supplementary Table S3** has been revised.

I would like it noted that histoclinical data based on small or very small cohorts represent the vast majority of the reports published nowadays, whereas the "concept" of presenting data on the largest possible cohort, which can be further confirmed and refined by analyzing an up-to-date dataset, is of high-value. I believe that reproducibility of histoclinical data patients is an important scientific "concept" that gives weight to our conclusions. *In fine*, when addressing a human illness it is of the highest relevance.

Another point I would like to make is that the analysis of large human cohorts is as relevant than results obtained in a single xenograft model. Although we are very happy to provide a new causal relationship, one has to wonder how the causal link established using a mice model would be more significant to human BC than rigorous correlative analyses of very large dataset of human patients. After two decades of accumulation of transcriptomic and genomic data performed by various labs and obtained from several thousands of patients from all origin, it is time to consider in-depth analyses of these data as significant *in vivo* results.

2) With the luminal/basal experiments (figure 2) do cells maintain luminal phenotype in collagen or acquire basal myoepithelial characteristics?

Based on their mesenchymal morphology, invasive phenotype in collagen, increased expression of vimentin and Snail1, along with the emergence of a new EpCAM^{low} and CD49^{low} "mesenchymal" population (**Fig.2b,d**), I would conclude that the luminal cells have gained a basal phenotype. A sentence has been added in the EMT paragraph of the Discussion to mention this point.

3) Is there evidence EFA6B directly interacts with cdc42? Do they have any proposed mechanisms for how this might work?

Since Cdc42 is activated in the absence of EFA6B, if EFA6B were to bind Cdc42 it would be to inhibit its activity either through a GAP (GTPase-activating protein) activity or by sequestration. Neither seems likely, because EFA6B does not bear a GAP activity, and because all GEFs are expressed at very low levels. Looking for a direct interaction of EFA6B with effector molecules by proteomic analysis from co-IP results or a 2- hybrid investigation did not uncover Cdc42 binding or that of its effectors.

In the presence of EFA6B, the inhibition of Cdc42 could be mediated through ARH-GAP molecules which display an Arf-GTP binding domain and a Rho/Rac/Cdc2 GAP domain, thus an ARH-GAP could link Arf6 activation to Cdc42 inhibition²². In which case, in the absence of EFA6B the decreased activation of Arf6 would lead to increased amount of activated Cdc42-GTP. Our candidate-based approach of this large family did not lead to conclusive results. At this point, we have to envision a more indirect pathway that we would very much like to discover. SiRNA screening of Cdc42-GEFs showed that several of them could regulate Cdc42 activation and collagen invasion of the EFA6B KO cells. It seems that Cdc42 activation is channeled through multiple signaling pathways making it a complex task to describe the molecular link between loss of EFA6B and Cdc42 activation.

A couple sentences mentioning these hypotheses have been added to the end of the CDC42 paragraph of the Discussion.

4) The authors describe in the text a more or less complete EMT, but KO 55 shows persistent E-cad expression. Is this more hybrid or partial EMT? Does Ecad localize to intercellular contacts in their model in EFA6B KOs- supporting a collective invasion phenotype. If not E-cad what is holding cells together, N- cad?

In **Supplementary Fig.3a**, we had shown confocal immunofluorescence images of E and N-cadherin staining in MCF10A WT and EFA6B KO55 cells. The E-cadherin staining is decreased while the N-cadherin staining is increased in the KO55 cells compared to WT cells. Further, it shows that E-cadherin is present as intracellular clusters, while N-cadherin is found at cell-cell contact in KO55 cells. Arrowheads have been added to the figure to indicate these characteristics. As shown in the panel **Supplementary Fig.3b**, we show that the cell-cell contacts mediated by the N-cadherin containing cell-cell junctions in KO55 cells are weaker, which indeed is believed to favor collective invasion.

“Partial EMT” would suggest that there is a continuum along a linear EMT program. I would rather say that it is hybrid, in the sense that the EMT observed, which would need to be refined, is most likely somehow specific to the loss of EFA6B and cell model dependent. In fact, in the HMLE cells the situation is different as the outcome of the EMT did not lead to a cadherin switch; the cells still express E-cadherin and did not up-regulate N-cadherin. The EMT is not a direct effect of EFA6B depletion but rather an indirect one caused by a global change, to which I would attribute a main part to the change in the environment. In any case, there are now plenty of data indicating that, at least in the mammary epithelium during normal development and cancer, the EMT can occur

without loss of E-cadherin^{23,24}.

A sentence has been added in the EMT paragraph of the Discussion indicating that the various EMT phenotypes observed across the MCF10A clones could reflect indirect scenarios of EMT induction.

5) At several points the author says “invalidated” whereas it seems to mean “knocked out”

We have made the requested corrections

6) The authors should provide more detail on rationale for MCF10A model which normally fails to form tight junctions in standard tissue conditions- if EFA6B is a tight junction regulator, why is MCF10A a good model to study its impact?

The criteria used for choosing the cell lines to assess the impact of knocking-out EFA6B were: 1) a **mammary** model since our results and others (Drs H. Sabe and G. Scitta) were pointing towards a role for EFA6B in breast cancer; 2) from **human** origin for comparison and relevance purposes with human breast cancer; 3) a normal **non-transformed** cells so as to not be “polluted” by a combination of effects together with those of transforming oncogene or tumor suppressor. In addition, because we had evidence that the loss of EFA6B was involved in early steps of tumorigenesis that we could only grasp in a normal cell population.

In fine, that meant studying **normal human mammary** cell lines. Therefore, we chose the MCF10A as it is the best characterized, and the HMLE because it contains all epithelial populations. I do not know of any normal human mammary cell line that has conserved the capacity to form tight junction in standard and 3D-collagen culture condition. The MCF10A95 from Dr. M. Balda²⁵ would have been an alternative but I did not think that they were sufficiently characterized. Thus, MCF10A and HMLE appeared to be the best choice.

EFA6B is not just a tight junction regulator. We had originally discovered that EFA6A stabilizes the tight junction through its role on the apical actin cytoskeleton. However, EFA6B has a more general effect on the EMT status of epithelial cells among which is regulation of the tight junction. The results presented in this study indicate that the role of EFA6B extends far beyond the tight junction. Indeed, the sole disruption of the tight junction, either by knock-down or knock-out of one of its components (See work from Tsukita’s lab and colleagues), or the general cell-cell junctions disruption by calcium-depletion in normal epithelial cells, or TGF exposure have never been shown to promote acquisition of invasion properties²⁶.

An introductory sentence explaining the rationale for using the MCF10A cells has been added in the first paragraph of the Results section.

7) In the legends they report N=3 average SEM, but they do not report the number of aggregates counted per condition per replicate.

In the Methods section, it was indicated that 100 aggregates per replicate were counted. The same information is now added in the figure legends as well.

8) given their prior paper showing association of EFA6B with claudin-low and aggressive TNBC, it might be interesting to focus Fig 8 on specifically TNBC IDC cases, in their

comparisons.

As requested by the Reviewer, we have added the comparison of *PSD4* mRNA expression between DCIS and IDC in the TN subtype, and between primary tumors and paired metastases (**Supplementary Fig.7b,c,d**). Interestingly, we observed the same results; higher expression in DCIS than found in IDC, and higher expression in primary tumors than found in metastases.

A new **Supplementary Fig.7** has been added. We have added the following sentence: “*The same comparative analysis of PSD4 expression between DCIS and IDC, then between primary tumors and paired metastases in the TN subtype showed similar results (Supplementary Fig.7), with higher expression in DCIS than in IDC, then in primary tumors than in metastases.*”

Reviewer #2 (Remarks to the Author)

This is an interesting manuscript describing how EFA6B suppresses an invasive phenotype in breast cancer. There are some important mechanistic observations: inhibiting this molecule leads to transcriptional reprogramming to support invasion. Collective invasion is controlled via regulation of ECM interactions and via CDC42-MRCK-Myosin activation. Patient data supports this mechanism. Before publication, a few comments need to be addressed:

1.What happens if you inhibit other collagenases apart from MMP14- like MMP2/MMP13, etc?

We have focused on MMP14 because 1) with MMP2 it is the most effective collagen type I protease *in vitro*²⁷; 2) our proteases' inhibitors profiling pointed towards MMP14 and alike; 3) we observed a focused degradation of the gelatin suggesting the implication of a membrane-associated protease concentrated in invadopodia rather than a soluble protease; and 4) breast cancer *in vivo* models are strongly indicating a major role for MMP14^{28,29}.

We have looked at MMP2 and MMP13 as suggested by the Reviewer. MMP-13 is not expressed in MCF10A WT or EFA6B KO55 cells. Please see **Figure for Reviewers 1a**. MMP2 is equally expressed in MCF10A WT and EFA6B KO cells. siRNA-mediated down-regulation of MMP2 to our surprise accelerated the invasion process. Please see **Figure for Reviewers 1b**. To measure the stimulation effect the assay was done in 3mg/ml collagen and quantified earlier at 36hrs. Note that siMMP2 had no effect on MCF10 WT cells invasive properties and that the siMMP2-stimulated invasion in EFA6B KO cells is abrogated by siMMP14 (data not shown). This intriguing observation does not bring key information to our current work. Thus, it is provided to the Reviewers, but not included in the manuscript.

At this point, without totally excluding the role of another protease, considering the potent effect of its down-modulation and concentration in invadopodia, MMP14 appears as a key protease of our invasive model.

2. Better measurement of myosin light chain activity is needed (blot in figure 6 not convincing). Quantification of the wbs is recommended (and statistical analysis), and authors should use immuno- fluorescence to measure phospho-myosin and where is located in the invasive structures.

We have repeated the anti-pMLC WB with the MCF10A KO55 and KO2 cell lines and quantified all

our WBs. The increase though moderate is very robust in both cells lines. The results are shown in **Fig.6b** (KO55) and **Supplementary Fig.6e** (KO2).

We have carried out confocal immunofluorescence using the dually phosphorylated pMLC2 (pMLC) specific antibody (Cell Signalling Technology #3674). The results are shown in the **Figure for Reviewers 1c**. We could not measure a significant difference in the whole anti-pMLC fluorescent signal, which I would attribute to a lower quantitative sensitivity of the IF compared to WB, especially that the levels of fluorescence intensity are already quite high in MCF10A WT cells. In addition, we could not observe a clear co-localization nor enrichment within or at the vicinity of the invadopodia, which might be due to the low amount of pMLC present at any time in these structures and/or the high turnover of phosphorylation/dephosphorylation. Also, the overall increased of pMLC appears to be distributed all over the cell and not only in invadopodia. Indeed, we observed that in MCF10A WT cells the pMLC is primarily visible in the perinuclear area. In KO cells, its distribution was enriched at the cell periphery where it colocalized with F-actin cables. The fact that the EFA6B KO cells display an augmentation of the total amount of pMLC and a cell redistribution of pMLC in F-actin cables reminiscent of contractile structures is in support of a higher contractile activity of the KOEFA6B cells measured in our in-gel contractility assay (**Fig.6a**).

3. What happens after ko of RhoC? And its GTPase activity? This small GTPase plays a key role in metastasis. Or RhoA+RhoC combined depletion?

We have measured the levels of activated RhoC using the pull-down assay. We did not find a significant difference compared to the WT cells (**Fig.6h**). Using siRNA we have repressed the expression of RhoC or both RhoC and RhoA in the KO55 cells. In neither case, did we observe an effect on the invasive capacities of the cells (**Fig.6e** and **Supplementary Fig.5**).

4. Quantification of pull downs needs to be provided, how many times where these pull downs performed? Statistical analysis is needed.

Each pull-down assays were performed at least 3 times in independent experiments as it was indicated in the Methods section and now in the figure legends. Quantification of the WB and statistical analyses were provided in the text of the Results section and are now directly in the figures as well. We have added the results for RhoC. Significant and reproducible activation was only observed for Cdc42.

5. In general: in the paper there are no measurements of actual invasion just protrusive aggregates? "After 48h, protrusions were quantified using a phase contrast microscope by counting 100 cellular aggregates per well. Cells with at least one membrane extension of at least 2 microns' length were considered invasive". This is not an invasion assay per se. This just quantifies protrusions, which could be inefficient protrusions and could not lead to actual invasion. Authors need to measure cell/group of cells displacement over time (the whole cell needs to invade not just make a protrusion).

I certainly agree with the Reviewer that looking solely at membrane protrusions is not an adequate assay to assess invasive properties. We used this assay only after having determined that the

cells were invasive by observing invasion in 3D collagen overtime up to 7 days (**Fig.1c, Fig.6c**), and that the cells could degrade gelatin (**Fig.3a**) and fibrillary collagen in 3D (**Fig.3c**). Invasion was further demonstrated by showing that it was dependent on MMP14 and the formation of invadopodia. After we made the observation that EFA6B KO cells were invasive in 3D collagen gels we searched for an assay that would be quantitative, robust, fast and easy. 48h post-seeding in 3D collagen, spheroids with long protrusions of at least 2 μ m was determined to represent a reliable assessment of invasion in our cell model. We have an excellent correlation between the formation of the long protrusions seen after 2 days and the formation of large invasive networks after 7 days as presented in the introductory **Fig.1**. As mentioned in my response to Reviewer#1 (major comment 2.), we have observed thousands of cell aggregates of all our clones over long periods, as we always keep our samples for extra days. We are confident that the quantitative assay we are presenting is a reliable proxy to evaluate the invasive capacities of our cell lines.

In support to all our assays of invasion, we are now presenting time-lapse imaging of invasion in the **Supplementary Movies 1 and 2**. Please refer to our comments to Reviewer#1's major comment 2.

We hope that the Reviewers will agree that our data are in strong support of collective invasion induced upon EFA6B depletion.

6. Figure 6A, Rho and Rac ko give an intermediate phenotype regarding contractility, not as strong as Cdc42, but there is an effect. Please discuss. Can authors show the phenotype in all assays after Rac and RhoA/C ko? Including cell morphology, invadopodia and cell invasion?

As indicated in response to comment 3, we are now showing images illustrating the phenotypes of cells plated on collagen, as well as the invadopodia labeling, and of the 3D-cell aggregates grown in fibrillary collagen matrix (**Supplementary Fig.5**).

We have noticed that the effects of the various siRNA on the cell morphology were milder when the cells were grown on a thick collagen layer as compared to directly on glass coverslips. In addition, the cells formed numerous and long filopodia when plated on collagen. We observed that siRhoA induced the formation of short thick F-actin cables oriented in all directions and that siRhoC rounded up the cells and induced the formation of F-actin filaments surrounding the periphery of the cells and often the nucleus. The combination of both siRNA against RhoA and RhoC further exacerbated the formation of the short thick F-actin cables along with appearance of dorsal spikes. In contrast, siRac1 elongated the cells and induced the formation of disorganized thin F-actin filaments forming a random meshwork. SiCdc42 increased the size of the cells including their height, the amount of cortical F-actin cables while reducing the amount of filopodia. In addition, siCdc42 caused a dispersion of the cortactin staining in small dots throughout the cells. While all the small G proteins affected the F-actin organization, only the depletion of Cdc42 had an effect on the formation of the invadopodia visible as large orange spots where F-actin (red) and (cortactin) are co-enriched.

This was confirmed by looking at the invasive behavior of the cells grown in 3D collagen. Indeed, all cell aggregates invaded the collagen except for those transfected with the siCdc42 that remained as round aggregates similar to those of MCF10A WT cells. Compared to the MCF10A KO55, siRhoA, siRhoC or siRhoA plus siRhoC aggregates produced long plasma membrane protrusions (indicated by arrowheads), while siRac1 aggregates were reorganized to form

branched single cell-wide elongated chains.

Regarding Rac1, we agree with the Reviewer that the decrease in Rac1 has some little effect on contractility (similarly to RhoA) and even on early invasion, yet not on long term invasion (**new Supplementary Fig.5**). In fact, Rac1-depleted cells display a slightly slower invasion rate reflected by a lower number of protrusions measured at day 2. However, we could not detect any activation of Rac1 in either of our EFA6B KO cell lines. We are not excluding some minor contribution of Rac1 but by no means comparable to Cdc42 whose depletion is the only one to display such potent inhibitory effect on invasion. A sentence has been added in the Discussion section to comment on that point.

Reviewer #3 (Remarks to the Author):

EFA6A, B, C and D are guanine nucleotide exchange factors that activate the small GTPase ARF6. One of these, EFA6B, has been shown to promote cell-cell contact in epithelial cells, and to restore epithelial morphology to moderately transformed cells. Here the authors present evidence that loss of the EFA6B gene (PSD4) in a weakly transformed breast epithelial cell line, MCF10A, leads to epithelial/mesenchymal transition (EMT) and collective invasion of the cells into a 3D collagen matrix. The data indicate that loss of EFA6B triggers transcriptional upregulation of EMT-inducing genes including SNAIL1, TWIST1 and ZEB1, a change in integrin repertoire and a downregulation of epithelial-specific genes encoding tight junction and cell-cell adhesion proteins. This transition was accompanied by increased activation of Cdc42 and formation of MMP14-enriched invadopodia, which are necessary for penetration of the collagen matrix. Importantly, patient-derived data indicate that EFA6B expression is reduced in invasive breast carcinoma, and that invasive tumors display transcriptional signatures similar to those of EFA6B-deficient MCF10A cells.

Overall, the study is well conceived and carefully conducted. In general, the data are convincing and support the conclusions drawn by the authors. The inclusion of clinical samples and correlation with outcomes is important and strengthens the impact of the study. There are, however, several issues that need to be addressed:

1. All of the imaging data needs to be quantified. The authors' claims of colocalization appear over- interpreted in some cases, and would require unbiased quantitative analysis in any case. This is particularly notable in Fig. 4E, where the authors claim that b1-integrin colocalizes with MMP14 in EFA6B-deficient cells, but not control MCF10A. This is clearly not the case, based on the images shown.

I do admit that our set of images were not of the highest quality. We have resumed from scratch by making new cell populations expressing homogeneously MMP14-mCherry at low levels. To perform the quantification of the co-localization we have designed a process whereby all experiments and samples were imaged following the exact same procedure detailed in the Methods section. Briefly, all pictures were taken with the same settings, and collected at the very bottom of the cells where invadopodia are located. Co-localization was measured using the ImageJ plug-in JACoP³⁰ and controlled with the Costes randomization test. The number of

repeats (n=3), total cells analyzed and statistical analyses have been indicated in the figure legends. New images illustrating the co-localization of cortactin and ITG β 1 with MMP14-mCherry are now included along with the quantitation in **Fig.3J-k** and **Fig.4e-h**, respectively. We hope that the Reviewer will find this new set of data convincing and in support our conclusion that invadopodia formed upon EFA6 depletion are ITG β 1-based and enriched in MMP14.

2. Similarly, none of the immunoblots are quantified. All blots should be representative of at least 3 independent experiments and quantified to demonstrate statistical significance. This is particularly important for the blots showing activation of Cdc42 but not Rac1 or RhoA, but should be done for all blots.

As mentioned in my response to Reviewer#2 (comment 4), the quantification of the pull-down assays was presented in the Results and Methods sections. Taking in account that it was not visible enough the numbers are now indicated directly below the WB. In order not to overload **Fig.5**, the quantification of all the WBs is shown in the form of bar graphs in the **Supplementary Fig.3f**. We thank the reviewers for prompting us to perform a thorough statistical analysis of our data, which consistently supported our conclusions.

3. A related point – it is somewhat surprising that Cdc42 is the only Rho family GTPase activated under these conditions. Is this a property of 3D culture in collagen, or does it also occur in 2D if cells are plated on collagen-coated plastic?

The experiments were indeed conducted with cells grown on collagen-coated plastic as it requires very large amounts of cells. Each pull-down requires a minimum of 20×10^6 cells and up to 80×10^6 depending on the small G protein. I would certainly agree that it would be best to get the information from invading cells but this is just not practically feasible from cells embedded in collagen. We have now stated clearly in the Methods section that the experiments were carried out using cells grown on collagen-coated plastic.

We are convinced that Cdc42 is highly activated in our EFA6B KO cells. However, considering the relative low sensitivity of the pull-down assay we would not totally exclude that low amounts of other small G proteins are activated locally and/or very transiently and somehow contribute to the invasive process in 3D. This point has been added at the beginning of the CDC42 paragraph of the Discussion.

4. On p4, the authors state that the transition from E-cadherin to N-cadherin, which occurs during EMT, was confirmed by RT-qPCR, but in the next paragraph state that they “did not notice a change in E- or N- cadherin expression”. What does this mean? That there was a transcriptional change without a corresponding change in protein expression?

I believe the Reviewer is referring to the first sentence of the penultimate paragraph p4 (**Fig.5**): “Further analysis by RT-qPCR confirmed the E/N-cadherin switch, ...” and the second sentence of the following paragraph of the same page: “Although we did not notice a change in E- or N-cadherin expression, ...”. Indeed, the first sentence refers to the results obtained in MCF10A EFA6B KO cells where E-/N-cadherin switch occurred, but the second sentence referred to the HMLE cell clones where the E-/N-cadherin switch was not observed. This is one of the reasons why we went

on to analyze the EMT-TFs profile expression as it better defines EMT.

We have rephrase the second sentence as follows: *In contrast to MCF10A EFA6B KO cells, we did not notice a change in E- or N-cadherin expression. However, we found a strong increase of vimentin and a slight but consistent decrease of CLDN3 expression.*

5. In referring to Fig. 1D, the authors state that EFA6B-deficient cells exhibited more protrusions than controls, but that this was not accompanied by “migratory properties”. What does this mean? Isn’t collective invasion considered migration?

I agree that the wording was confusing by linking inappropriately the formation of the invasive protrusions and cell motility. The second half of the sentence, that is now separated, was solely meant to indicate that the loss of EFA6B expression did not increase the migratory capacities of the cells analyzed in 1D wound- healing or single-cell tracking assays.

Collective invasion indeed is migration but our point was to say that although the cells move through the fibrillary collagen effectively it is not due to or associated with an increase of the cell motility capacities.

6. Presumably many of the downstream effects of EFA6B loss are due to reduced activation of ARF6 at specific cellular locations. How do the authors think this leads to EMT, activation of Cdc42 and formation of invadopodia?

EFA6B, as most regulators of small G proteins, is a protein presenting multiple functional domains from which the Sec7 catalytic Arf6-activating domain is only one of them. The best-studied C-terminal domain is known to interact with various effectors such as actin, actinin or β -arrestin³¹⁻³³. There is little known about the large (about half of the molecule) and most divergent N-terminal domain among the EFA6 molecules, and even less about the PH domain that seems to play other functions than simply associating EFA6 molecules to PIP2 enriched lipid membranes³⁴. Splicing variants for all four EFA6 members have been described for which we know nothing^{18,35-37}, except for an EFA6A short splicing variant that lacks both the N-terminus and catalytic Sec7 domain. Of note, this PH-Cterminus variant is functional and induces dendrite branching in neuronal cells³⁸. Thus, it is more than likely that the absence of EFA6B has pleiotropic effects far beyond the sole decrease of Arf6 activation. Further, the latter is not totally absent in EFA6B KO cells, most likely because of the presence of other Arf6-GEFs.

Regarding the link between the loss of EFA6B and EMT, and the loss of EFA6B and Cdc42 activation, please refer to my answers to Reviewer#1 (major point 4, and minor point 3).

The relation between Cdc42, invadopodia and invasion has been well documented and was mentioned in the Discussion section. It is still possible that besides Cdc42, other factors including some that might be more specific to the EFA6/Arf6 pathway, contribute to the formation of degradative invadopodia.

However, at this point our data are supportive of the implication of a pathway predominantly Cdc42-dependent. Among all the molecules that we have knocked-down by siRNA, the down-regulation of Cdc42 was by far the most effective at blocking invadopodia and invasion.

Thank you for your consideration and patience.

Sincerely,

Frédéric Luton, PhD.
Research Director,
Inserm

Bibliography

1. Hu, M. *et al.* Regulation of in situ to invasive breast carcinoma transition. *Cancer Cell* **13**, 394–406 (2008).
2. Behbod, F. *et al.* An intraductal human-in-mouse transplantation model mimics the subtypes of ductal carcinoma in situ. *Breast Cancer Res* **11**, R66 (2009).
3. Russell, T. D. *et al.* Myoepithelial cell differentiation markers in ductal carcinoma in situ progression. *Am J Pathol* **185**, 3076–3089 (2015).
4. Lodillinsky, C. *et al.* p63/MT1-MMP axis is required for in situ to invasive transition in basal-like breast cancer. *Oncogene* **35**, 344–357 (2016).
5. Castagnino, A. *et al.* Coronin 1C promotes triple-negative breast cancer invasiveness through regulation of MT1-MMP traffic and invadopodia function. *Oncogene* **37**, 6425–6441 (2018).
6. Balda, M. S. & Matter, K. Tight junctions and the regulation of gene expression. *Biochim Biophys Acta* **1788**, 761–7 (2009).
7. González-Mariscal, L. *et al.* Tight junctions and the regulation of gene expression. *Semin Cell Dev Biol* **36**, 213–223 (2014).
8. Spadaro, D., Tapia, R., Pulimeno, P. & Citi, S. The control of gene expression and cell proliferation by the epithelial apical junctional complex. *Essays Biochem* **53**, 83–93 (2012).
9. Tsukita, K., Yano, T., Tamura, A. & Tsukita, S. Reciprocal Association between the Apical Junctional Complex and AMPK: A Promising Therapeutic Target for Epithelial/Endothelial Barrier Function? *Int J Mol Sci* **20**, (2019).
10. Citi, S., Guerrero, D., Spadaro, D. & Shah, J. Epithelial junctions and Rho family GTPases: the zonular signalosome. *Small GTPases* **5**, 1–15 (2014).

11. Melzer, C., Hass, R., von der Ohe, J., Lehnert, H. & Ungefroren, H. The role of TGF- β and its crosstalk with RAC1/RAC1b signaling in breast and pancreas carcinoma. *Cell Commun Signal* **15**, 19 (2017).
12. Imamura, T., Hikita, A. & Inoue, Y. The roles of TGF- β signaling in carcinogenesis and breast cancer metastasis. *Breast Cancer* **19**, 118–124 (2012).
13. Hardy, K. M., Booth, B. W., Hendrix, M. J. C., Salomon, D. S. & Strizzi, L. ErbB/EGF signaling and EMT in mammary development and breast cancer. *J Mammary Gland Biol Neoplasia* **15**, 191–199 (2010).
14. Guo, S. & Deng, C.-X. Effect of Stromal Cells in Tumor Microenvironment on Metastasis Initiation. *Int J Biol Sci* **14**, 2083–2093 (2018).
15. Kai, F., Drain, A. P. & Weaver, V. M. The Extracellular Matrix Modulates the Metastatic Journey. *Developmental Cell* **49**, 332–346 (2019).
16. Zangari, J. *et al.* EFA6B antagonizes breast cancer. *Cancer Res* **74**, 5493–506 (2014).
17. Theard, D. *et al.* USP9x-mediated deubiquitination of EFA6 regulates de novo tight junction assembly. *Embo J* **29**, 1499–509 (2010).
18. Luton, F. *et al.* EFA6, exchange factor for ARF6, regulates the actin cytoskeleton and associated tight junction in response to E-cadherin engagement. *Mol Biol Cell* **15**, 1134–45 (2004).
19. Klein, S., Partisani, M., Franco, M. & Luton, F. EFA6 facilitates the assembly of the tight junction by coordinating an Arf6-dependent and -independent pathway. *J Biol Chem* **283**, 30129–38 (2008).
20. Lee, S. *et al.* Differentially expressed genes regulating the progression of ductal carcinoma in situ to invasive breast cancer. *Cancer Res.* **72**, 4574–4586 (2012).
21. Knudsen, E. S. *et al.* Progression of ductal carcinoma in situ to invasive breast cancer is associated with gene expression programs of EMT and myoepithelia. *Breast Cancer Res. Treat.* **133**, 1009–1024 (2012).
22. Dubois, T. *et al.* Golgi-localized GAP for Cdc42 functions downstream of ARF1 to control Arp2/3 complex and F-actin dynamics. *Nat Cell Biol* **7**, 353–364 (2005).
23. Shamir, E. R. & Ewald, A. J. Adhesion in mammary development: novel roles for E-cadherin in individual and collective cell migration. *Curr Top Dev Biol* **112**, 353–382 (2015).
24. Padmanaban, V. *et al.* E-cadherin is required for metastasis in multiple models of breast cancer. *Nature* **573**, 439–444 (2019).
25. Sourisseau, T. *et al.* Regulation of PCNA and cyclin D1 expression and epithelial morphogenesis by the ZO-1-regulated transcription factor ZONAB/DbpA. *Mol Cell Biol* **26**, 2387–2398 (2006).
26. Ozdamar, B. *et al.* Regulation of the polarity protein Par6 by TGFbeta receptors controls epithelial cell plasticity. *Science* **307**, 1603–9 (2005).
27. Amar, S., Smith, L. & Fields, G. B. Matrix metalloproteinase collagenolysis in health and disease. *Biochim Biophys Acta Mol Cell Res* **1864**, 1940–1951 (2017).
28. Willis, A. L., Sabeh, F., Li, X.-Y. & Weiss, S. J. Extracellular matrix determinants and the

- regulation of cancer cell invasion stratagems. *J Microsc* **251**, 250–260 (2013).
29. Feinberg, T. Y. *et al.* Divergent Matrix-Remodeling Strategies Distinguish Developmental from Neoplastic Mammary Epithelial Cell Invasion Programs. *Dev Cell* **47**, 145–160.e6 (2018).
 30. Bolte, S. & Cordelières, F. P. A guided tour into subcellular colocalization analysis in light microscopy. *J Microsc* **224**, 213–232 (2006).
 31. Macia, E. *et al.* The C-terminal domain of EFA6A interacts directly with F-actin and assembles F- actin bundles. *Sci Rep* **9**, 19209 (2019).
 32. Milanini, J. *et al.* EFA6 proteins regulate lumen formation through α -actinin 1. *J Cell Sci* **131**, (2018).
 33. Macia, E., Partisani, M., Paleotti, O., Luton, F. & Franco, M. Arf6 negatively controls the rapid recycling of the beta2 adrenergic receptor. *J Cell Sci* **125**, 4026–35 (2012).
 34. Macia, E. *et al.* The pleckstrin homology domain of the Arf6-specific exchange factor EFA6 localizes to the plasma membrane by interacting with phosphatidylinositol 4,5-bisphosphate and F-actin. *J Biol Chem* **283**, 19836–44 (2008).
 35. Derrien, V. *et al.* A conserved C-terminal domain of EFA6-family ARF6-guanine nucleotide exchange factors induces lengthening of microvilli-like membrane protrusions. *J. Cell Sci.* **115**, 2867–2879 (2002).
 36. Matsuya, S. *et al.* Cellular and subcellular localization of EFA6C, a third member of the EFA6 family, in adult mouse Purkinje cells. *J Neurochem* **93**, 674–85 (2005).
 37. Fukaya, M., Ohta, S., Hara, Y., Tamaki, H. & Sakagami, H. Distinct subcellular localization of alternative splicing variants of EFA6D, a guanine nucleotide exchange factor for Arf6, in the mouse brain. *J Comp Neurol* **524**, 2531–2552 (2016).
 38. Sironi, C. *et al.* EFA6A encodes two isoforms with distinct biological activities in neuronal cells. *J Cell Sci* **122**, 2108–18 (2009).

REVIEWERS' COMMENTS

Reviewer #1 (Remarks to the Author):

Racha and colleagues have improved their manuscript with new experiments and textual revisions that have strengthened the study. The re-analysis and time-lapse movies are now very convincing for collective invasion. They also very convincingly show that EFA6B loss has different effects on EMT factors and target genes depending on cell line model, but that the ultimate phenotype of collective invasion is shared between models. This suggests there is an EFA6B depending pathway upstream of the specific form of EMT that modulates collective invasion induction. In addition, the dcis.com experiments are also important for supporting the in vivo functional relevance of EFA6B levels for invasive transition and bring another dimension to their study. These revisions have substantially improved the work but there are a few minor technical issues that would be important to address before publication.

1) The authors should shore up their quantitative analyses for Fig. 8 using the data they have already collected. They report percentage p63 in the text but I could not find a table or figure supporting this statement. This data should be provided for the reader. It is also not clear what is the unit of replication they are talking about in the text regarding % +/- sd: tumors, biological replicates, fields? It also doesn't seem clear from the images in Fig. 8a that differences in % of p63+ cells are all that different at weeks 3 and 4. If so, the conclusions should be qualified more carefully.

2) Likewise, the authors should provide quantitative support for their claim of earlier basement degradation/invasion past basement membrane in the KO dcis.com line. The supplemental Figure 7a arrowheads are not clear what they were highlighting. The arrowhead in the EFA6B 2 week panel appears to be pointing to inside a tumor nest.

3) The authors suggest that EFA6B KO in dcis.com is associated with increased invasion and basal differentiation which is interesting. But in light of their data showing that KO of HMLE and MCF10A induce EMT phenotypes, what is effect of EFA6B KO on collective invasion and EMT status of DCIS.COM tumors? What is their Ecad status? Do they show increased expression of other basal markers? Alternatively, the authors should clarify that underlying mechanisms for increased invasion in DCIS.COM model with EFA6B KO remain to be worked out.

4) The supplemental movies are helpful, but could use some improved clarity. In Movie 1, both WT and control guide are shown but only WT is labeled. Likewise, the different KOs have different replicate movies. It would be better to label each movie with a title, as it was hard to follow. Likewise for Movie 2, why are low power movies shown for the KOs but not for WT (and/or control guide)?

5) "Further analysis by RT-qPCR confirmed the E/N-cadherin switch, the decrease of TJ markers and of CK14 whose down-regulation was recently shown to mark an advanced mesenchymal state in melanoma" ref- 18. This was a study by Blanpain and colleagues looking at mammary breast cancer models- not melanoma.

Reviewer #2 (Remarks to the Author):

Authors have addressed all my comments. I believe that the data authors include for the reviewer only, should be included in the published version of the manuscript-as it is very informative.

Reviewer #3 (Remarks to the Author):

The authors have done an excellent job of addressing my earlier concerns, and in my opinion, those of the other reviewers. Significant new data have been added that address my concerns about quantitation.

Reviewer #1 (Remarks to the Author):

Racha and colleagues have improved their manuscript with new experiments and textual revisions that have strengthened the study. The re-analysis and time-lapse movies are now very convincing for collective invasion. They also very convincingly show that EFA6B loss has different effects on EMT factors and target genes depending on cell line model, but that the ultimate phenotype of collective invasion is shared between models. This suggests there is an EFA6B depending pathway upstream of the specific form of EMT that modulates collective invasion induction. In addition, the dcis.com experiments are also important for supporting the in vivo functional relevance of EFA6B levels for invasive transition and bring another dimension to their study. These revisions have substantially improved the work but there are a few minor technical issues that would be important to address before publication.

1) The authors should shore up their quantitative analyses for Fig. 8 using the data they have already collected. They report percentage p63 in the text but I could not find a table or figure supporting this statement. This data should be provided for the reader. It is also not clear what is the unit of replication they are talking about in the text regarding % +/- sd: tumors, biological replicates, fields? It also doesn't seem clear from the images in Fig. 8a that differences in % of p63+ cells are all that different at weeks 3 and 4. If so, the conclusions should be qualified more carefully.

A figure showing the quantification of p63-positive cells at week 2 and 3 along with its legend was presented in Supplementary Figure 7. We are now presenting the quantification up to 5 weeks. Indeed the difference in p63⁺ cells is highest at week 2 and decreases over time.

2) Likewise, the authors should provide quantitative support for their claim of earlier basement degradation/invasion past basement membrane in the KO dcis.com line. The supplemental Figure 7a arrowheads are not clear what they were highlighting. The arrowhead in the EFA6B 2 week panel appears to be pointing to inside a tumor nest.

An additional graph is presented in Supplementary Figure 7 reporting the quantification of infiltrating foci clearly detected earlier in DCIS.com KOEFA6B tumors compared to DCIS.com WT tumors as indicated by: the loss of the monolayer organization of p63⁺ cells, the remodeling of the stroma and the fusion of the tumor foci which is a sign of invasion and thus basement membrane degradation. The arrows do not mark the absence of the basement membrane that can not be ascertained on 2D sections, however they point to infiltrating foci which likely breach through the basement membrane. The term "basement degradation" has been replaced by "tumor infiltration" to better match the observations.

3) The authors suggest that EFA6B KO in dcis.com is associated with increased invasion and basal differentiation which is interesting. But in light of their data showing that KO of HMLE and MCF10A induce EMT phenotypes, what is effect of EFA6B KO on collective invasion and EMT status of DCIS.COM tumors? What is their Ecad status? Do they show increased expression of other basal markers? Alternatively, the authors should clarify that underlying mechanisms for increased invasion in DCIS.COM model with EFA6B KO remain to be worked out.

I agree with the Reviewer, we have not defined the EMT status of the DCIS.com KOEFA6B cells nor analyze their invasive behavior in 3D collagen matrix. Our statement was essentially based on the p63 marker and the somehow more elongated cell morphology observed in the tumors. We have tuned down our interpretation of the images and stated that further work is needed to define the invasive mechanism of the DCIS.com KOEFA6B cells in vitro.

4) *The supplemental movies are helpful, but could use some improved clarity. In Movie 1, both WT and control guide are shown but only WT is labeled. Likewise, the different KOs have different replicate movies. It would be better to label each movie with a title, as it was hard to follow. Likewise for Movie 2, why are low power movies shown for the KOs but not for WT (and/or control guide)?*

The Supplemental movies are now individually labeled. Low magnification movies are only shown for the KOEFA6B spheroids as they expand away from the spheroid, unlike the WT that remain compact and are better seen at higher magnification.

5) "Further analysis by RT-qPCR confirmed the E/N-cadherin switch, the decrease of TJ markers and of CK14 whose down-regulation was recently shown to mark an advanced mesenchymal state in melanoma" ref- 18. This was a study by Blanpain and colleagues looking at mammary breast cancer models- not melanoma.

The Reviewer is right but since the main study had been conducted by analyzing a skin squamous cell carcinoma (SCC) model we had not referred to their extended results on breast models. This has now been corrected by writing "... advanced mesenchymal state in melanoma and breast tumors".

Reviewer #2 (Remarks to the Author):

Authors have addressed all my comments. ***I believe that the data authors include for the reviewer only, should be included in the published version of the manuscript-as it is very informative.***

The MMP21/MMP13 results have been included in the Supplementary Figure 2 and the pMLC localization in Supplementary Figure 5.

Reviewer #3 (Remarks to the Author):

The authors have done an excellent job of addressing my earlier concerns, and in my opinion, those of the other reviewers. Significant new data have been added that address my concerns about quantitation.